# SPINDOC binds PARP1 to facilitate PARylation

Fen Yang[1,2], Jianji Chen[1,3], Bin Liu[1], Guozhen Gao [1], Manu Sebastian[1], Collene Jeter[1], Jianjun Shen[1], Maria D. Person [4] & Mark T. Bedford [1✉]

SPINDOC is tightly associated with the histone H3K4me3 effector protein SPIN1. To gain a better understanding of the biological roles of SPINDOC, we identified its interacting proteins. Unexpectedly, SPINDOC forms two mutually exclusive protein complexes, one with SPIN1 and the other with PARP1. Consistent with its ability to directly interact with PARP1, SPIN-DOC expression is induced by DNA damage, likely by KLF4, and recruited to DNA lesions with dynamics that follows PARP1. In SPINDOC knockout cells, the levels of PARylation are reduced, in both the absence and presence of DNA damage. The SPINDOC/PARP1 interaction promotes the clearance of PARP1 from damaged DNA, and also impacts the expression of known transcriptional targets of PARP1. To address the in vivo roles of SPIN-DOC in PARP1 regulation, we generate SPINDOC knockout mice, which are viable, but slightly smaller than their wildtype counterparts. The KO mice display reduced levels of PARylation and, like PARP1 KO mice, are hypersensitive to IR-induced DNA damage. The findings identify a SPIN1-independent role for SPINDOC in the regulation of PARP1-mediated PARylation and the DNA damage response.

[1] Department of Epigenetics and Molecular Carcinogenesis, The University of Texas MD Anderson Cancer Center, Smithville, TX 78957, USA. [2] Department of Biochemistry and Molecular Biology, School of Basic Medical Sciences, Nanjing Medical University, Nanjing 211166, China. [3] Graduate Program in Genetics & Epigenetics, The University of Texas MD Anderson Cancer Center UTHealth Graduate School of Biomedical Sciences, Houston, TX 77030, USA. [4] Center for Biomedical Research Support The University of Texas at Austin, Austin, TX 78712, USA. ✉email: mtbedford@mdanderson.org

Spindlin1 (SPIN1) is a known histone-code effector protein, which is composed almost solely of three Tudor domains. With its second Tudor domain, it reads the H3K4me3 mark, which is the canonical signal for transcriptional activation. It also has the ability to recognize combinatorial marks like H3R8me2a and H3K9me3 through its first Tudor domain, when they occur in the presence of K3K4me3[1–5]. We identified SPINDOC as a stable component of the SPIN1 protein complex[6], and this finding has subsequently been independently validated[7]. SPINDOC refers to its ability to dock with SPIN1, which was previously called C11orf84 before it was functionally deorphanized. The interaction surfaces between SPIN1 and SPINDOC were mapped using a novel serial-capture affinity-purification approach in conjunction with cross-linking mass spectrometry[8]. This mapping study identified numerous contacts between SPINDOC and SPIN1, including intermolecular cross-links within each of the three Tudor domains of SPIN1. However, recent structural studies have revealed that it is the Tudor-3 domain that is critical for the recognition of SPINDOC[9].

To gain a better understanding of how SPINDOC may work, we tagged it and purified its protein complex. Previous studies have focused on purifying the tagged SPIN1 complex[6–8], so this is the reciprocal experiment. As expected, we identified SPIN1 as the major SPINDOC interactor. We also identified a similar number of peptides that corresponded to poly(ADP-ribose) polymerase (PARP1). Further analysis revealed that SPINDOC did not function as a linker to recruit PARP1 to SPIN1. Rather, SPINDOC forms two distinct protein complexes: one with SPIN1 and a second with PARP1. This study will focus on the characterization of the significance of the latter complex.

PARP1 is the best-characterized member of the diphtheria toxin-like ADP-ribosyl transferases (ARTDs) family of proteins, which has 18 ARTDs. Only PARP-1, PARP-2, and PARP-3 are activated by DNA-strand breaks, while PARP-1 is responsible for ~90% of the global poly-ADP-ribosylation (PARylation) synthesis following DNA-strand breakage[10,11]. Early work implicated PARP-1 in DNA repair by demonstrating that PARP-1-deficient mice are highly sensitive to γ-irradiation and DNA-damaging agents[12,13]. PARP1 plays critical roles in the repair of single-strand breaks (SSBs) and DNA double-strand breaks (DSBs)[14]. PARP1 is able to bind DSBs and modulates recruitment of DSB-repair factors that are important for homologous recombination (HR) as well as nonhomologous end joining (NHEJ)[15]. Moreover, PARP1 interacts with and stimulates the activity of multiple DNA replication-associated proteins and is itself activated in response to replicative stress, as observed by increased PARylation in hydroxyurea-treated cells and cells with unligated Okazaki fragments[16,17]. Sensing DNA lesion, PARP1 triggers the DNA-damage response (DDR) process via two independent mechanisms: (1) by PARylating its substrates, which helps recruit DDR proteins that harbor PAR-binding domains[14], and (2) by promoting the transcription of genes that are critical for a DDR[18]. PARP1 can facilitate nucleosome disassembly by specifically PARylating lysine residues of the core histone tails, including H3K27, H3K37, H2AK13, H2BK30, and H4K16, which results in chromatin relaxation[19]. When PARP1 is complexed with histone PARylation factor 1 (HPF1), it is able to PARylate serine residues on histone[20]. In addition, the linker histone H1.2 undergoes rapid PARP1-dependent chromatin dissociation through PARylation of its C terminus and further proteasomal degradation upon DNA damage[21]. PARP1 has been shown to regulate a number of different transcriptional pathways, likely by modifying and activating p300[22]. For example, PARP1 inhibition resulted in diminished radiation-induced NF-κB binding to target gene loci[23,24]. PARP1 interacts with nuclear respiratory factor 1 (NRF1) and plays a role in NRF1 transcriptional regulation[25].

PARP1 inhibition was found to cause increased occupancy of the E2F4/P130-repressive complex on BRCA1 and RAD51 promoters[26]. PARP1 occupies the human sodium-iodide symporter (hNIS) promoter, and inhibition of PARP1 enzymatic activity results in increased hNIS reporter activity as well as endogenous hNIS mRNA expression[27]. PARP1 also regulates RNA processing, like alternative splicing, RNA modification, mRNA stability, and mRNA translation[28,29].

In the present study, we find that SPINDOC is distributed between two different protein complexes, one that harbors SPIN1 and the other that harbors PARP1. SPINDOC mRNA and protein levels are elevated upon DNA damage. SPINDOC facilitates PARP1-mediated PARylation and downstream DDR-associated gene transcription. SPINDOC-deficient mice are more ionizing radiation (IR) sensitive than wild-type counterparts. Cumulatively, our data indicate that SPINDOC is a key auxiliary factor for a PARP1-mediating DDR.

## Results

**SPINDOC forms two independent protein complexes: one with PARP1 and the other with SPIN1.** SPINDOC was identified as a SPIN1 family-interacting protein, which masked the transcriptional coactivator functions of SPIN1 by blocking its ability to "read" the H3K4me3 activator mark[6]. We speculated that SPINDOC would function as a cofactor of the SPIN1 family that recruits other chromatin-associated proteins. Thus, we purified the ectopically expressed GFP-SPINDOC protein complex from HEK293T cells and identified the components of this complex, using a traditional immunoprecipitation (IP), followed by SDS-PAGE, in-gel tryptic digestion, and mass spectrometry (MS) analysis. As expected, we found that SPIN1 selectively interacted with GFP-SPINDOC, but not with the pEGFP-C1 empty vector control. We note that GFP-SPINDOC can enrich for SPINDOC itself, possible due to homodimerization, but there is also some binding of the GFP control to endogenous SPINDOC. In addition, we observed a SPINDOC-specific interaction with PARP1 as well as with a number of histones (Fig. 1a, Supplementary dataset 1). These data raised the intriguing possibility that SPIN1 recruits SPINDOC, which in turn recruits PARP1. Alternatively, cells may harbor distinct SPIN1/SPINDOC and PARP1/SPINDOC complexes.

To test these scenarios, we again purified the GFP-SPINDOC protein complex, but this time using a construct with a 43-amino-acid deletion in SPINDOC (Δ251–293aa), which we have mapped as the region required for the interaction with SPIN1, and confirmed by others[9]. Again, we observed an interaction with PARP1, but not with SPIN1, as predicted (Fig. 1b, Supplementary dataset 2). This indicated that SPINDOC and PARP1 can complex in the absence of SPIN1. Next, we isolated the endogenous SPIN1 protein complex from both wild-type and CRISPR-generated SPINDOC-knockout cells (Supplementary Fig. 1). In the absence of SPINDOC, PARP1 did not purify with SPIN1 (Fig. 1c, Supplementary dataset 3). Thus, these three independent MS experiments showed that SPINDOC can bind PARP1, independent of SPIN1, raising the possibility of the existence of two independent SPINDOC complexes, one with PARP1 and a second with SPIN1. To independently verify the SPINDOC–PARP1 interaction, we carried out co-IP experiments using ectopically expressed GFP-SPINDOC and Flag-PARP1 in HEK293T cells, and we found that SPINDOC could bind both PARP1 and SPIN1, while PARP1 only bound to SPINDOC (Fig. 1d).

Next, we confirmed that SPINDOC indeed splits its presence between two different complexes. To do this, we performed two independent 15–35% glycerol-gradient sedimentation experiments

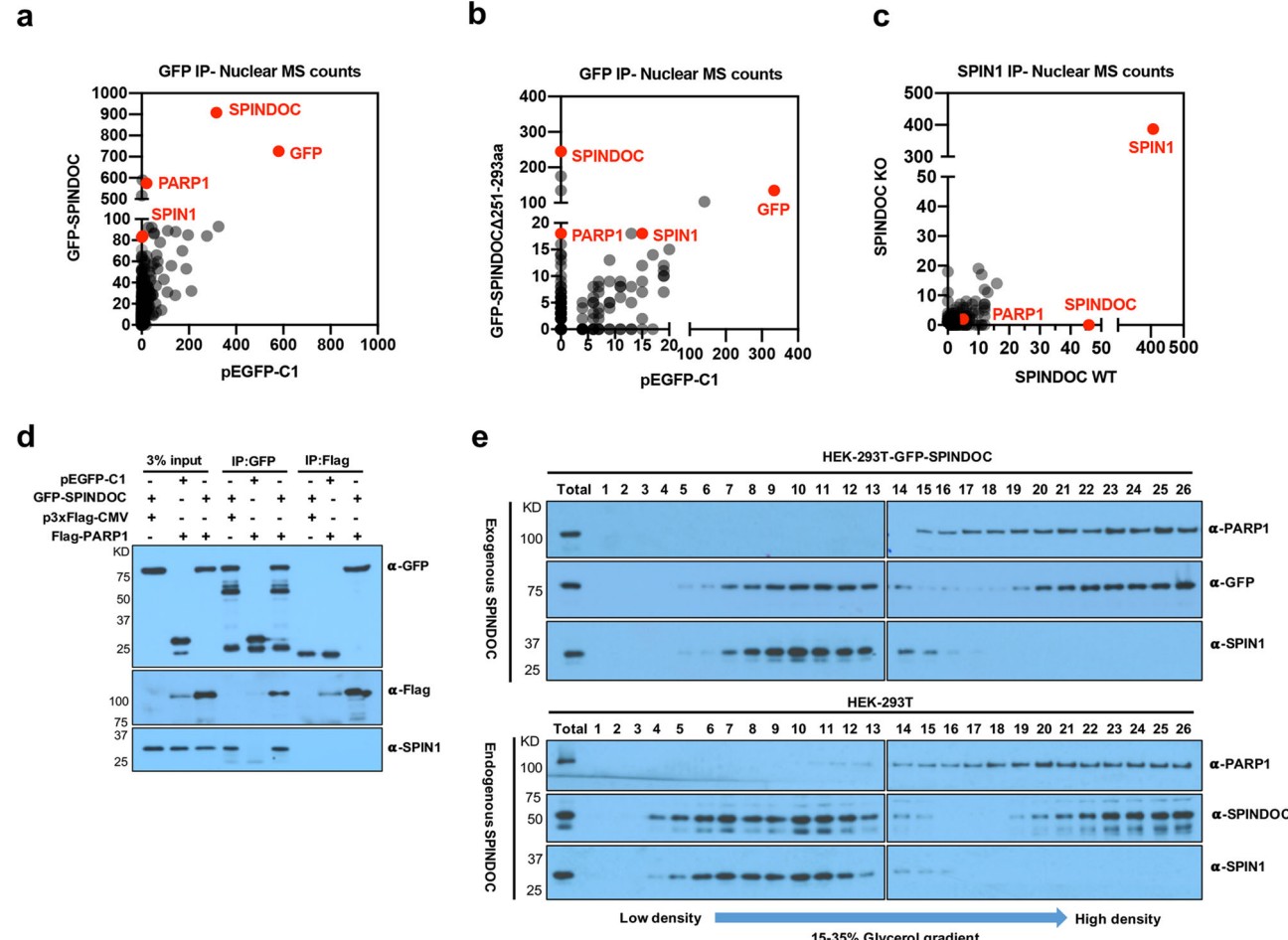

**Fig. 1 SPINDOC is involved in two different complexes. a** HEK293T cells were transfected with pEGFP-C1 vector and GFP-SPINDOC, 48 h later, cells were harvested and processed into cytoplasmic and nuclear fractions. Each fraction was immunoprecipitated by anti-GFP antibodies and subjected to MS analysis. Peptide spectral counts are presented as a scatterplot, and the empty vector group serves as a control. **b** HEK293T cells were transfected with pEGFP-C1 vector and GFP-SPINDOCΔ251–293aa, and the same procedure as followed as in (**a**). **c** SPIN1 IP was performed from HEK293T-SPINDOC WT and KO cell lines and then analyzed by MS. **d** HEK293T cells were transfected with the indicated vectors, and incubated for 48 h. GFP IPs and Flag IPs were performed to evaluate the interactions between SPINDOC, PARP1, and SPIN1. The results were repeated three times independent experiments, shown as representative blots. **e** Lysates from HEK293T cells and HEK293T cells transfected with GFP-SPINDOC were centrifuged with 15~35% glycerol gradients and divided into 26 fractions from top to bottom (density from low to high). All samples were analyzed by Western blot, the "Total" is the original lysate prior to gradient centrifugation. The results were repeated three times independent experiments, shown as representative blots.

to probe the composition of protein complexes isolated from HEK293T cells that overexpressed GFP-tagged SPINDOC (Fig. 1e —upper panels), and also from HEK293T cells that only expressed endogenous SPINDOC (Fig. 1e—lower panels). In both cases, SPINDOC was split into two nonoverlapping complexes.

**Mapping the regions of interaction between SPINDOC and PARP1.** Next, we mapped the regions of SPINDOC and PARP1 that were required for their interaction. Initially, HEK293T cells were transfected with six different truncated versions of Flag-PARP1 and a mutant in the PARP1-cleavage site (D214N), together with the full-length SPINDOC fused to GFP. We then performed an anti-GFP IP and a Western blot with anti-Flag to observe the co-IP potential of the different deletion and mutant constructs, where the pEGFP-C1 vector alone served as a negative control. The result demonstrated that the PARP1 DNA-binding domain (DBD) fragment was responsible for interacting with SPINDOC (Fig. 2a, b). We next focused on mapping the region of SPINDOC that bound PARP1. To do this, we generated seven deletion constructs of GFP-fused SPINDOC, which were each cotransfected with Flag-PARP1 into HEK293T cells and subjected

to an anti-GFP IP. The data revealed that the SPINDOC 115–165aa region interacted with PARP1 (Fig. 2c, d). Cotransfection of the 3xFlag-CMV vector alone served as a negative control. Thus, these mapping experiments identified the GFP-SPINDOC 115–165aa and Flag-PARP1-DBD as the interaction surfaces of these two proteins. This raised the possibility that the binding of PARP1 to DNA may regulate the PARP1/SPINDOC interaction.

To investigate this possibility, we cotransfected GFP-SPINDOC 115–165aa and Flag-PARP1-DBD, and then performed an anti-GFP IP, in the presence or absence of DNase I and ethidium bromide (EtBr). The result showed that the SPINDOC 115–165aa fragment indeed interacted with PARP1-DBD, and this interaction was stronger when the cellular DNA was digested or treated with an intercalator (Supplementary Fig. 2a, b), which suggested that DNA and SPINDOC compete for the same binding region within the N terminus of PARP1. To further confirm the interaction of SPINDOC and PARP1 that can be impacted by the presence of DNA, we performed an in vitro GST pulldown assay using purified GST and GST-SPINDOC coincubated with recombinant PARP1, in the presence or absence of sheared

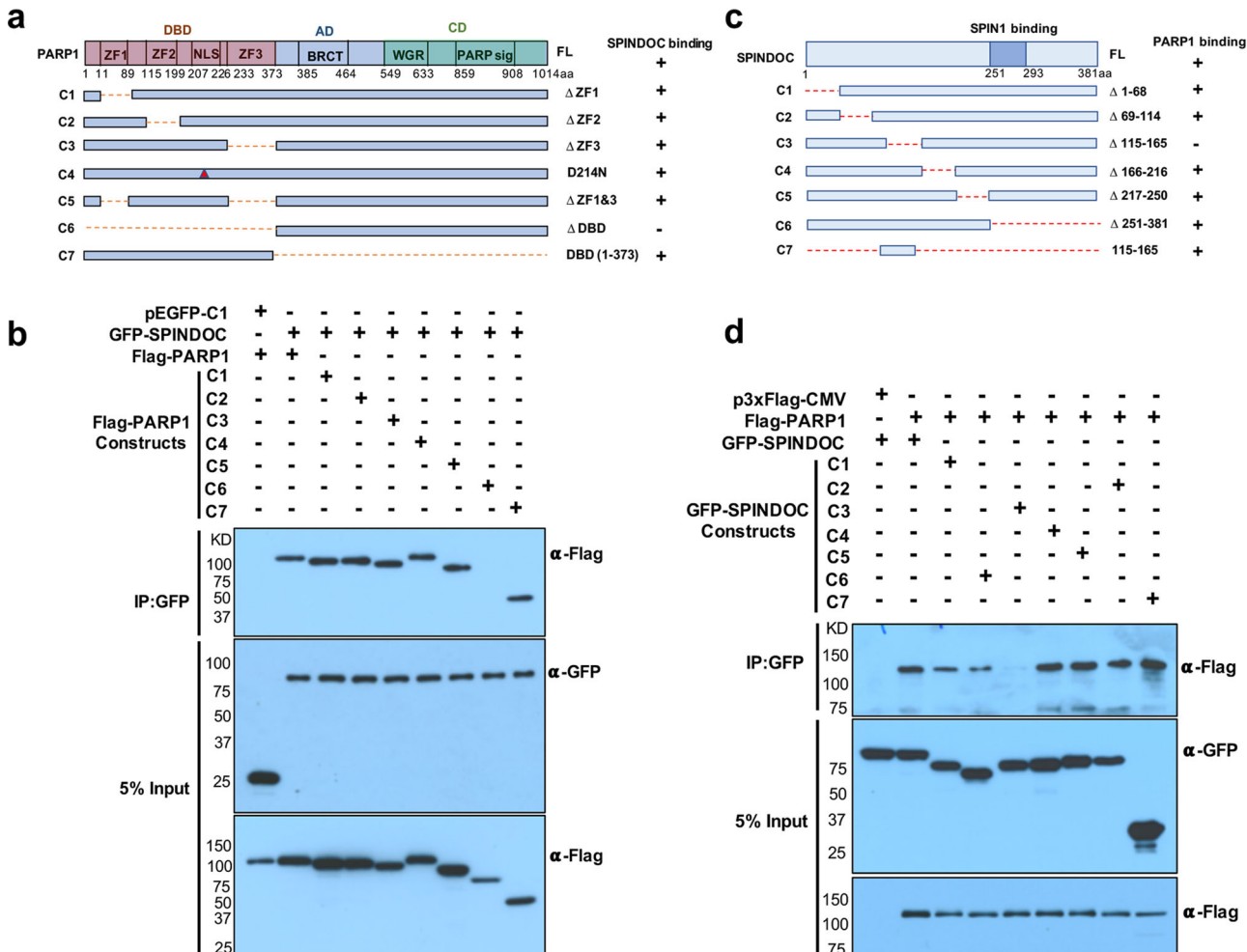

**Fig. 2 Mapping the SPINDOC and PARP1-interaction domain. a** Domain structure and deletion/mutation constructs for PARP1. Up panel: The major domains and active catalytic sites of PARP1 are marked, DBD: DNA-binding domain, including 1–373aa, ZF1: zinc finger 1, ZF2: zinc finger 2, ZF3: zinc finger 3, NLS: nuclear-localization signal, D214N: caspase-cleavage site, BRCT: BRCA1 C-terminal, AD: automodification domain, CD: catalytic domains, including WGR domain and PARP1-signature domain. **b** HEK293T cells were transfected with the listed constructs and subjected to GFP IP after 48 h, and analyzed by Western blot using the indicated antibodies. The results were repeated three times independent experiments, shown as representative blots. **c** Domain structure and deletion constructs for SPINDOC. SPINDOC251–293aa is SPIN1-binding motif. **d** HEK293T cells were transfected with the listed constructs and subjected to GFP IP after 48 h, and analyzed by Western blot using the indicated antibodies. The results were shown as representative blots from three times independent experiments.

DNA. The data demonstrated that SPINDOC interacts with PARP1 directly and that this binding is decreased in the presence of a DNA competitor (Supplementary Fig. 2c). Moreover, coexpression of Flag-PARP1 with GFP-SPINDOC results in increased protein levels of PARP1 based on Flag signal (Fig. 1d and Fig. 2b). We thus performed a Flag-SPINDOC transfection, using increasing amounts of Flag-SPINDOC plasmid DNA, to investigate the effects of SPINDOC overexpression on endogenous PARP1 levels. We found that high SPINDOC level had a minor effect on enhancing endogenous PARP1 stability (Supplementary Fig. 2d).

PARP1 and PARP2 have complementary functions in the surveillance and maintenance of genome integrity in both deficient cellular and animal models[30,31]. We further investigated whether SPINDOC interacted with PARP2 using GFP-SPINDOC IP followed by Western blot for PARP2. PARP2 did not co-IP with GFP-SPINDOC (Supplementary Fig. 2e), which indicates that SPINDOC is specific for interaction with PARP1. This finding is perhaps not surprising because the DNA-binding domains of PARP1 and PARP2 are very distinct.

**Transcriptome analysis reveals a link between SPINDOC and the DDR pathway.** To further investigate the function of SPIN-DOC, we used CRISPR technology to generate SPINDOC KO in HEK293T and Hela cell lines. Three different pairs of CRISPR sgRNAs were used to target exons 1, 2, and 6 of SPINDOC (Supplementary Fig. 1a), and the deletions caused a loss of SPINDOC protein in HEK293T (in 5 independent clones) and Hela cells (in eight independent clones) (Supplementary Fig. 1b, c). For subsequent studies using HEK293T cells, we selected Exon1–1# and Exon2–6# as SPINDOC KO cell lines KO-1 and KO-2, and Exon1–8# and Exon2–1# as the corresponding SPIN-DOC WT cell lines WT-1 and WT-2. For Hela cells, we selected Exon2–1# as the SPINDOC KO and Exon2–9# as the SPINDOC WT. A side-by-side analysis of the proliferation rates of these four HEK293T cell lines and two Hela cell lines revealed that the SPINDOC KO cell lines grew slower than their WT counterparts. For this experiment, we used 1000 (HEK293T) or 2000 (Hela) cells plated on a 96-well plate and maintained for five days, during which time relative confluence rates were measured using the Celigo (Supplementary Fig. 3a, b). Using the same two pairs of

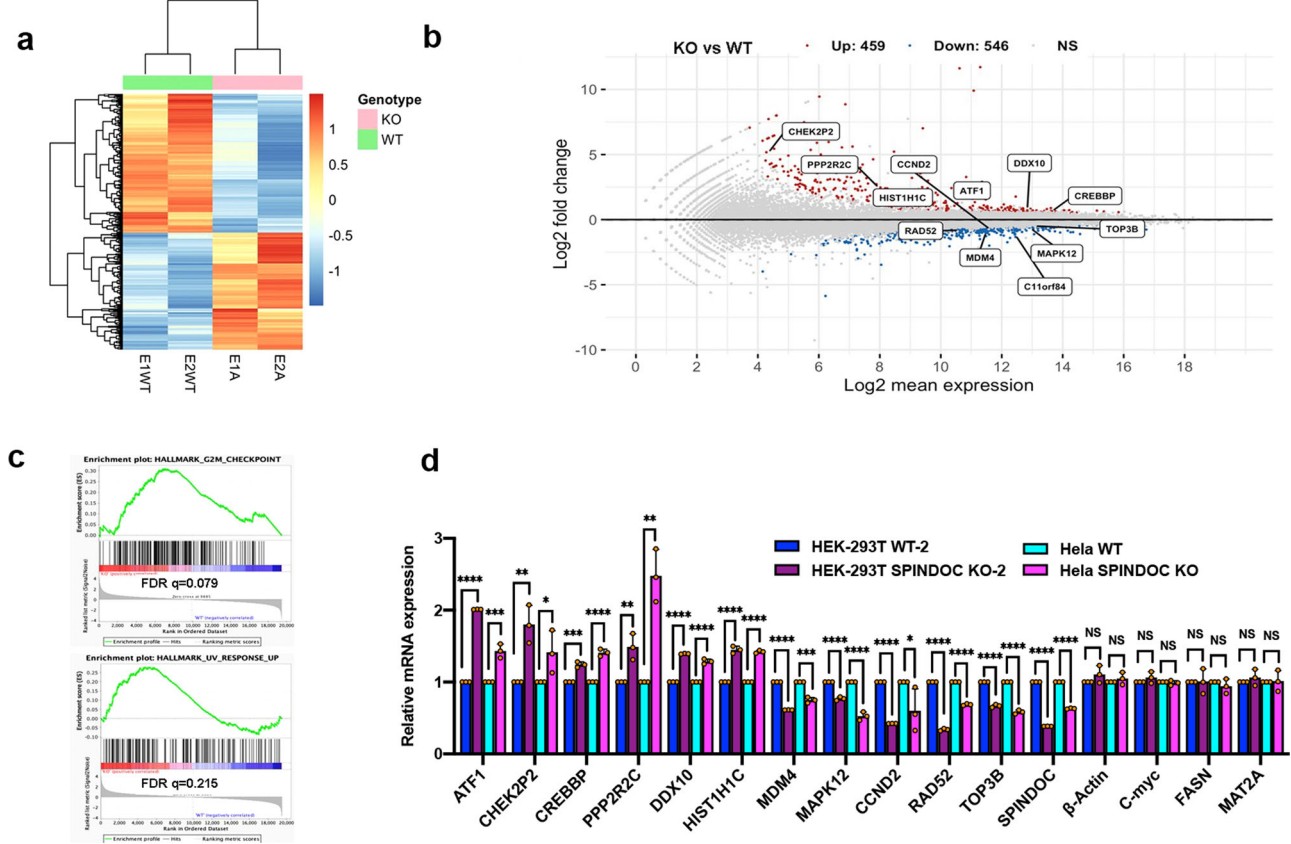

**Fig. 3 SPINDOC is associated with DDR as revealed by RNA sequencing and GESA analysis. a** Heatmap of differentially expressed genes in two independent pairs of HEK293T WT and KO cell lines that were used from RNA-sequencing analysis. **b** The MA plot shows the overlap of differentially expressed genes from the RNA-seq dataset. Red dots are upregulated genes (Up), blue dots are down-regulated genes (Down), gray dots are nonsignificant genes (NS), and marked genes are DNA-damage and repair-associated genes. **c** Differentially expressed gene sets (SPINDOC KO vs WT) were enriched for hallmark of "G2M Checkpoint" and "UV response UP" as established by GESA analysis. **d** Upregulated, downregulated and nondifferentiated genes were validated by RT-qPCR. Graphs represent mean ± SD, $n = 3$ biologically independent samples. Statistical analysis was performed using one-tailed Student's t-test with CHEK2P2: HEK293T, $P = 0.0032$, Hela, $P = 0.0367$; PPP2R2C: HEK293T, $P = 0.0049$, Hela, $P = 0.0011$; CCND2: Hela, $P = 0.0432$; ***$P < 0.001$, ****$P < 0.0001$, NS: nonsignificant.

HEK293T SPINDOC KO/WT cell lines, we carried out an RNA-sequencing (RNA-seq) analysis to investigate what specific transcripts and pathways were potentially regulated by SPINDOC. Comprehensive RNA-seq analysis demonstrated that 1005 genes were differentially regulated, including 459 up-regulated genes and 546 downregulated genes, which are represented in the heatmap (Fig. 3a) and a M (log ratio)–A (mean average) plot (Fig. 3b). In the MA plot, both upregulated transcripts (ATF1, CHEK2P2, CREBBP, PPP2R2C, DDX10, and HIST1H1C) and downregulated transcripts (MDM4, MAPK12, CCND2, RAD52, and TOP3B) are classified as DDR-associated genes (Fig. 3b). All differentiated genes are listed in Supplementary dataset 4. C11orf84 is the original name for SPINDOC, and as expected, it is listed in the downregulated gene cluster. All highlighted genes in Fig. 3b were validated by RT-qPCR (Fig. 3d) and Western blot (except for CHEK2P2) (Supplementary Fig. 3c). As controls, we also included a number of genes (β-Actin, FASN, MAT2A, and c-Myc) that did not display altered expression levels upon SPINDOC loss (Fig. 3d), and all expression changes are consistent with RNA-seq. At the protein level, reduced PARP1 expression is observed in SPINDOC KO cells (Supplementary Fig. 3c). Except for DDX10, all protein-level changes are consistent with mRNA-level changes (Supplementary Fig. 3c). Furthermore, these differentiated genes in SPINDOC KO vs WT were enriched in hallmark "G2M Checkpoint" (FDR, $q = 0.079$) and "UV response UP" (FDR, $q = 0.215$)

gene sets by GESA analysis (Fig. 3c, Supplementary dataset 5), which further indicated a potential role for SPINDOC in the DDR pathway. The RNA-seq dataset also revealed hallmark enrichment for "KRAS signaling", "TNFA signaling", and "apoptosis" (Supplementary Fig. 3d, Supplementary dataset 5). We need to bear in mind that the SPINDOC KO will likely impact the transcription-regulation properties of both the SPINDOC/SPIN1 complex and the SPINDOC/PARP1 complexes. For the purposes of this study, we will focus on the SPINDOC/PARP1 function and its link to DDR.

**SPINDOC expression is induced by DNA damage.** PARP1 is a DNA nick sensor that directly binds DNA lesions, and is a key component of the DNA-repair process[32]. Given that SPINDOC strongly interacted with PARP1, and that transcriptome analysis revealed a link between SPINDOC loss and altered expression of a gene signature for a DDR (Fig. 3c, Supplementary dataset 5), we next investigated whether SPINDOC itself could be impacted by DNA damage. To do this, we treated cells with the DSB-inducing agents Etoposide and X-ray IR, and we analyzed SPINDOC protein levels at a series of timepoints post recovery. For this experiment, we treated both HEK293T (Fig. 4a) and Hela (Supplementary Fig. 4a) cells to ensure that the response we saw was not due to an idiosyncrasy within a single-cell type. In both cell types, we observed a dramatic induction of SPINDOC protein

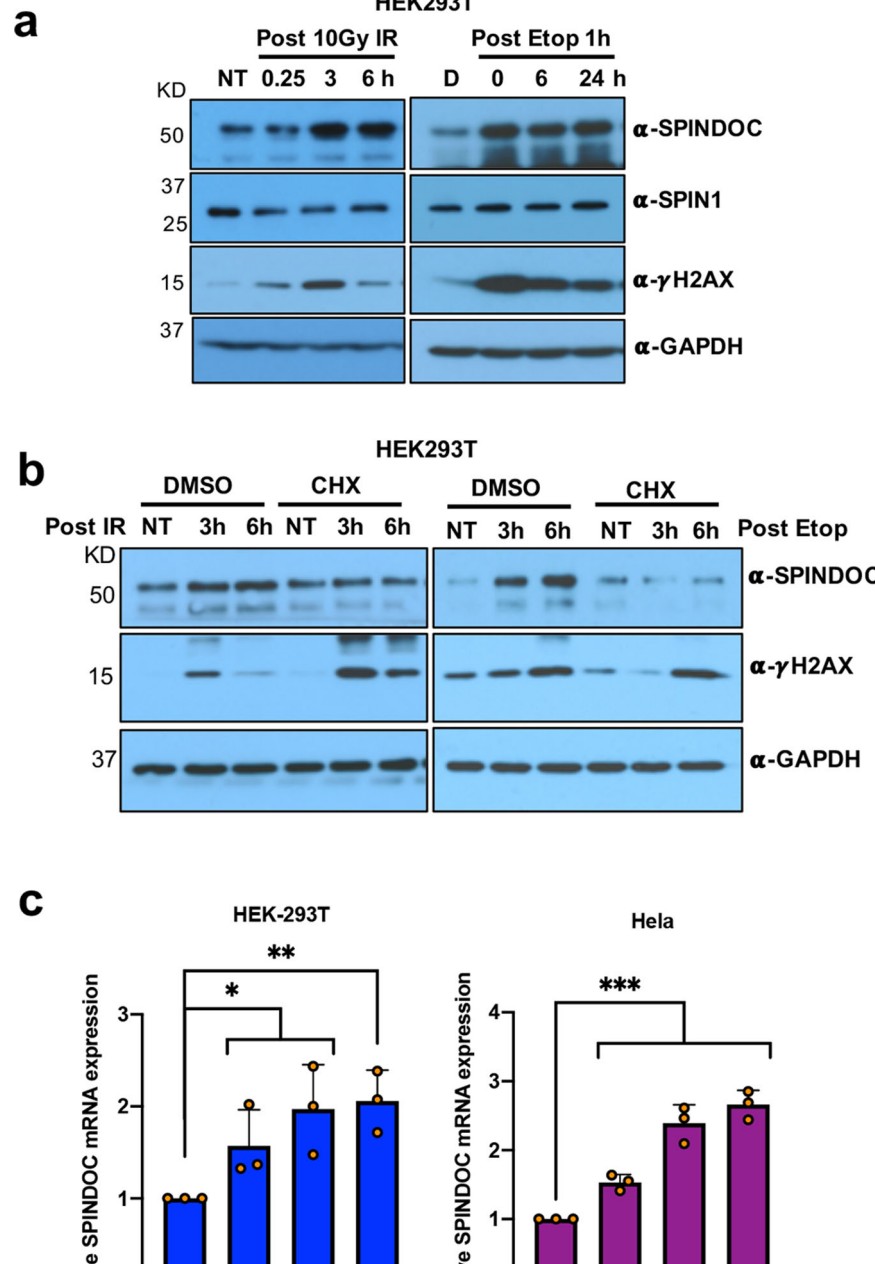

**Fig. 4 DSBs induce SPINDOC expression at both the mRNA and protein levels. a** HEK293T cells were treated with 10 Gy IR and 40 μM Etoposide for one hour before media replacement, and samples were taken at the indicated timepoints for Western blot analysis. Nontreated (NT) cells serve as a control, and DMSO treatment (D) as a solvent control. The results were shown as representative blots from three times independent experiments. **b** HEK293T cells were treated with DMSO or 10μM CHX and then immediately treated with 10-Gy IR (left panels) or 40μM Etoposide for an hour before replacing media (right panels) and incubated for either 3- or 6 h. Western blot analysis was then performed using the indicated antibodies. The results were repeated three times independent experiments, shown as representative blots. **c** HEK293T and Hela cells were treated with 10-Gy IR and cultured further for 0.5, 3, and 6 h. Total RNAs were extracted for RT-qPCR analysis of SPINDOC mRNA levels. Graphs represent mean ± SD, $n = 3$ biologically independent samples. Statistical analysis was performed using one-tailed Student's $t$-test with HEK293T IR treatment compared with NT, 0.5 h, $P = 0.0320$, 3 h, $P = 0.0126$, 6 h, $P = 0.0027$, ***$P < 0.001$.

levels when the cells were either Etoposide- or IR-treated. Furthermore, we performed reciprocal IPs for GFP-SPINDOC and PARP1 upon Etoposide treatment and the result showed that DNA damage dramatically increased the interaction of these two proteins (Supplementary Fig. 4b, c). This indicates that the SPINDOC/PARP1 interaction is promoted during DDR.

We next asked whether the DNA-damage-induced expression of SPINDOC occurred at the transcriptional level. To investigate this, HEK293T cells were treated with IR or Etoposide in the presence of cycloheximide (CHX), which is a protein-synthesis inhibitor in eukaryotes (Fig. 4b). In the presence of CHX, we did not observe the induction of SPINDOC protein levels after the

induction of either Etoposide- or IR-generated DNA damage, which suggests that the regulation is at the RNA level. To directly address this possibility, we next performed a RT-qPCR experiment that showed that the SPINDOC mRNA level steadily increased after IR exposure, in both HEK293T and Hela cells (Fig. 4c). Taken together, this suggests that SPINDOC mRNA expression is upregulated after DNA-damage, likely by a DNA damage-sensing transcription factor (TF) like p53[33], E2F1[34], NF-κB[35], or KLF4[36], to name a few.

**KLF4 regulates SPINDOC transcription in response to DNA damage.** The promoter region of SPINDOC was identified by the public website Ensembl and the UCSC genome browsers. We focused on the first 1000 bp upstream of the transcription-start site and performed a computational screen (http://jaspar.genereg.net/). We found that 58 putative TF-binding sites were predicted in this putative promoter region (Supplementary dataset 6). From this list, we selected three common TFs (STAT3[37], SP1[38], and KLF4[36]) that have been clearly implicated in a DDR. To investigate whether these TFs regulated the basal expression of SPINDOC, we knocked down each of the TFs independently, and found that KLF4 reduction resulted in close to a 50% decrease in SPINDOC-expression levels, which was the strongest effect among the three TFs tested (Supplementary Fig. 5a, b). We thus focused on the KLF4 as our potential TF candidate, which binds within 100 bp of the SPINDOC TSS (Fig. 5a). It has long been known that KLF4 is induced in response to DNA damage[39,40]. We could validate these reports in Hela cells, using both 10-Gy IR and 40 μM Etoposide as DNA-damaging agents (Fig. 5b). Furthermore, KLF4 knockdown blocked SPINDOC induction by Etoposide, at both the mRNA (Fig. 5c) and protein levels (Fig. 5d).

To confirm that the regulation of SPINDOC expression by KLF4 was direct, we cloned the 1000-bp SPINDOC promoter region that harbors the KLF4-binding motif (pGL3-SPINDOC-P), or a mutant form of this motif (pGL3-SPINDOC-P-Mut) into luciferase-reporter plasmid. We performed a series of dual-luciferase assays to evaluate the responsiveness of these two promoters. The wild-type promoter was mildly active on its own. This activity was enhanced by IR treatment, and cotransfection with HA-tagged KLF4, with the strongest activity seen with a combination of DNA damage and KLF4 overexpression. Conversely, the mutant form of the promoter was not activated by either KLF4 overexpression or IR treatment (Fig. 5e). Furthermore, we performed ChIP-qPCR assays to confirm that KLF4 binds to the predicted region (−95bp ~ −86bp) of the SPINDOC promoter, and IR treatment promoted the association of KLF4 with this genomic region (Fig. 5f). Taken together, SPINDOC transcription is induced by DSBs via KLF4 activation. These data indicate that KLF4 is likely the primary transcription factor responsible for driving the DNA-damage-induced expression of SPINDOC.

**SPINDOC colocalizes with PARP1 and regulates both PARP1 clearance and PARylation.** PARP1 acts as a highly sensitive sensor for DNA damage and is recruited rapidly to laser-induced DNA-damage tracks[41]. PARP1 produces PAR at newly generated DNA DSBs, which promotes local chromatin relaxation due to its negative charge and histone displacement, as well as facilitates the recruitment of repair factors, such as MRE11[42–44]. As a PARP1-interacting protein, SPINDOC might regulate some of these processes, and to investigate this possibility, we cotransfected Hela cells with GFP-SPINDOC and an RFP-αPARP1 nanobody with the presence or absence of PARP1 inhibitor PJ34 10μM treatment for an hour. These cells were then pulsed with Hoechst

and subjected to laser microirradiation-coupled live-cell imaging. In Fig. 6a, c, we show that both SPINDOC and PARP1 were quickly recruited to DNA lesions, after which PARP1 was cleared, but the SPINDOC signal remained strong even at the later time points. PARP1 is cleared within 300 s (Fig. 6a, b), which is in keeping with reports characterizing the RFP-PARP1 nanobody that was used in this study[45]. When cells were treated with PJ34 prior to pulsing with Hoechst, PARP1-recruitment efficiency was found to be reduced and its resident time extended (Fig. 6b, c). We also noticed that PARP1 fluorescence in the nontreated cells quickly returned to the very low level, which is consistent with its reported eviction by other proteins[46]. On the other hand, SPINDOC recruitment and resident time was not affected by deficient PARP1 activity (PJ34 treatment).

We next looked at PARP1 recruitment and resident time at sites of laser-induced damage, in the absence of SPINDOC. We should note here that Cas9 expression is present in the Hela SPINDOC KO cell line, as the cell line was generated using CRISPR/Cas9 technology. This may result in some degree of DDR activity in these KO cells, which could confound the interpretation of PARP1-recruitment dynamics presented here. That said, in WT cell line, PARP1 was released quickly, however, in Hela-SPINDOC KO cells, PARP1 clearance was markedly delayed (still present at 600 s). Importantly, this delayed release of PARP1 can be rescued by the reintroduction of SPINDOC into the SPINDOC KO cells, but not by the reexpression of mutant SPINDOC that cannot bind PARP1 (Fig. 6d, e). These findings could be indicative of reduced efficiency of DNA repair, possibly due to reduced PARP1 activity, in which case PARP1 needs to reside on DNA longer to fully execute its DNA-repair functions.

To test the link between SPINDOC and PARP1 activity, we treated SPINDOC WT and KO cell lines with IR (Fig. 6f) and Etoposide (Supplementary Fig. 6a), and we isolated lysates at a number of timepoints during recovery. In response to DNA damage, PARP1 is activated and known to undergo prominent auto-PARylation[14], and to modify a number of its substrates[19]. In the absence of SPINDOC, PARP1-mediated auto-PARylation is dramatically reduced, and with longer exposure, additional substrate PARylation is also observed to be diminished (Fig. 6f, Supplementary Fig. 6a). Clearly, SPINDOC not only regulates PARP1 protein levels, but also PARP1-mediated PARylation. To determine if this activation of PARP1 is direct, we performed an in vitro PARylation assay, using recombinant PARP1, and we found that GST-SPINDOC (but not GST) directly stimulated PARP1 enzymatic activity (Fig. 6g). Furthermore, a GST fusion harboring only the region that interacts with PARP1 (GST-SPINDOC115–165) is sufficient to induce this activation. To clarify that the regulation of PARP1, by SPINDOC, occurred at the protein level and not the RNA level, we performed qPCR for PARP1 under SPINDOC-knockout and -overexpression conditions, and found no impact on the transcriptional level (Fig. 6h).

Apart from the role of PARP1 in DNA-damage detection, it also has important functions in the regulation of gene expression as part of the subsequent repair response[47]. Thus, we next investigated the possible functions of SPINDOC in the context of gene regulation. We selected a panel of six genes that have been previously characterized as targets of PARP1 regulation, in response to DNA damage. These include positively regulated targets BRCA1, RAD51, NRF1 and NF-κB1 and negatively regulated genes P130 and hNIS[23–27]. In Hela SPINDOC WT cells, the transcription of PARP1 positively regulated targets (BRCA1, RAD51, NRF1, and NF-κB1) was induced by IR treatment, while the transcription of PARP1 negatively regulated genes (P130 and hNIS) was markedly decreased (Fig. 6i). The loss of SPINDOC resulted in varying degrees of attenuation of the regulation of this panel of genes, and included a general decrease in BRCA1

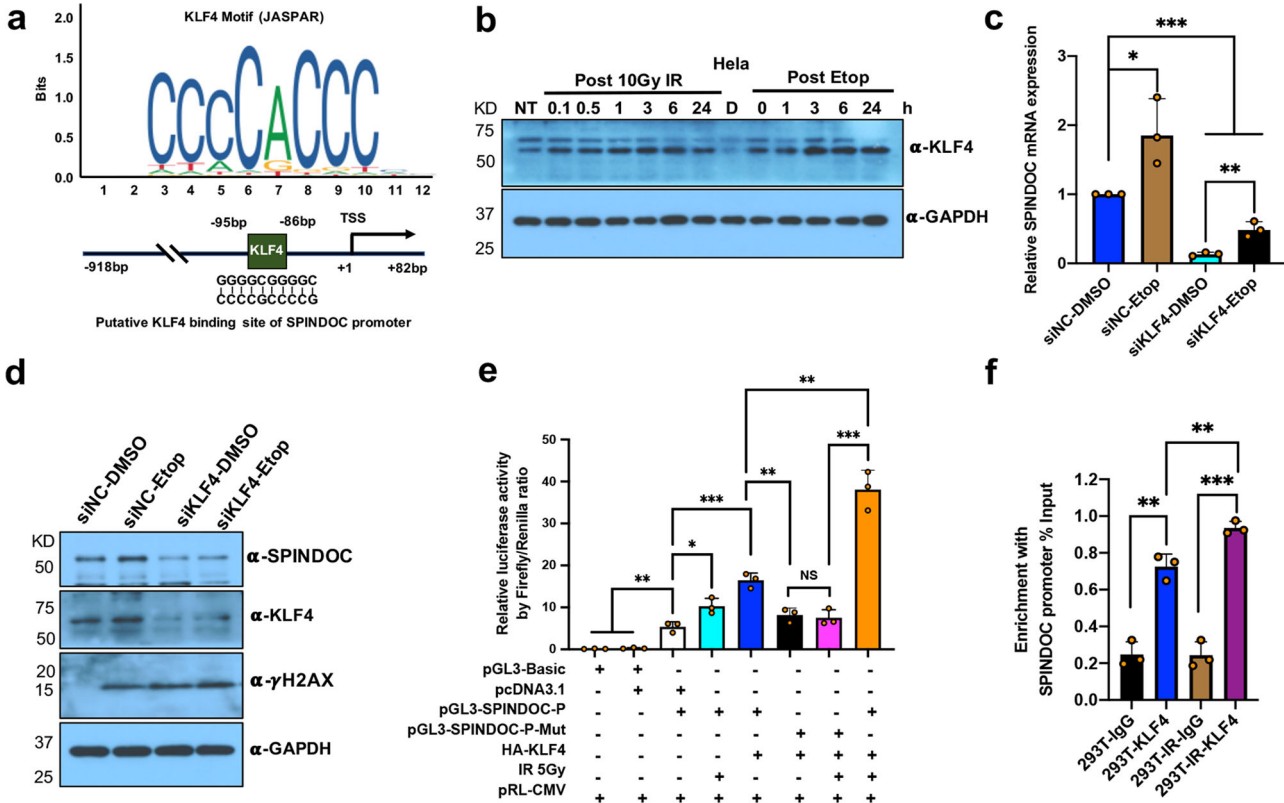

**Fig. 5 DSBs promote SPINDOC transcription by KLF4. a** Schematic showing the optimal KLF-binding motif from JASPAR website and the position of this KLF4-binding site in human SPINDOC promoter. **b** Hela cells were treated with 10-Gy IR and 40μM Etoposide for one hour before media replacement, and then incubated for the indicated timepoints and analyzed for anti-KLF4 expression by Western blot. The data are representatives of three independent experiments. **c** Hela cells were transfected with siNC or siKLF4 for 48 h and then treated with 40μM Etoposide or DMSO for one hour before media replacement, and then subjected to a further 3 h of recovery. Total RNAs were extracted for RT-qPCR to evaluate SPINDOC mRNA-level changes. Statistical analysis was performed using one-tailed Student's $t$-test. Error bas represent mean ± SD, $n = 3$ biologically independent experiments. *$P = 0.0164$, ** $P = 0.0026$, ***$P < 0.001$. **d** Hela cells were treated using the same procedure as in (**c**) and analyzed for SPINDOC and KLF4 protein levels by Western blot. **e** HEK293T cells were transfected with the indicated plasmids. pGL3-SPINDOC-P represents the cloned 1000-bp SPINDOC promoter depicted in (**a**). pGL3-SPINDOC-P-Mut harbors a deletion in the KLF4-binding site. At 48 h post-transfection, cells were treated with 5-Gy IR (as indicated) and then allowed 3 h for recovery. Cells were lysed and subjected to a dual-luciferase Firefly/Renilla assay. Error bas represent mean ± SD, $n = 3$ biologically independent experiments. Statistical analysis was performed using two-tailed Student's $t$-test. From left to right representing group 1–8, 3 vs 1, $P = 0.0017$; 3 vs 2, $P = 0.0019$; 4 vs 3, 0.0189; 5 vs 3, $P = 0.0008$; 6 vs 5, $P = 0.0040$; 7 vs 8, $P = 0.0005$; 7 vs 6, $P = 0.6733$; 8 vs 5, $P = 0.0016$. **f** Confluent 10-cm-dish HEK293T cells were treated with 10-Gy IR, and allowed to recover for 3 h. Cells were then subjected to ChIP-qPCR analysis using anti-KLF4 and anti-IgG antibodies. % Input was used to depict the relative KLF4 enrichment at the SPINDOC promoter. Graphs represent mean ± SD, $n = 3$ biologically independent samples. Statistical analysis was performed using two-tailed Student's $t$-test with KLF4 vs IgG, $P = 0.0010$; IR-KLF4 vs IR-IgG, $P = 0.0001$; NT-KLF4 vs IR-KLF4, $P = 0.0090$.

expression across all post-damage timepoints, no response in the case of NF-κB1, and a delayed response for RAD51 and NRF1. The opposite was seen for the two negatively regulated PARP1 targets, with both P130 and hNIS showing an increase in expression upon SPINDOC loss (Fig. 6i). Importantly, this altered pattern of response was not only observed in Hela SPINDOC KO cells, but also in HEK293 SPINDOC KO cells (Supplementary Fig. 6b). The altered gene expression patterns that we observed in SPINDOC KO cells were rescued with SPINDOC reexpression, in both Hela and HEK293T cell lines (Fig. 6j and Supplementary Fig. 6c). In addition, PARP1 overexpression also rescued the altered gene expression patterns seen in Hela SPINDOC KO cells, thus affirming these six genes as PARP1 targets (Supplementary Fig. 6e). To differentiate between PARP1 and SPIN1 complex-induced transcriptional changes, we rescued SPINDOC KO cells with pEGFP-C1, GFP-SPINDOC, GFP-SPINDOCΔ115–165aa (PARP-binding defect), and GFP-SPINDOCΔ251–293aa (SPIN1-binding defect) and treated with 10-Gy IR and recovery 0.5 h. The result showed that GFP-SPINDOC and GFP-

SPINDOCΔ251–293aa, but not GFP-SPINDOCΔ115–165aa and pEGFP-C1, could rescue the regulation of these PARP1 targets and SPIN1 does not play a role in this process (Supplementary Fig. 6d). We also find that SPINDOC KO cells display a reduced survival after DNA damage (Supplementary Fig. 6f), when compared with wild-type cells. Thus, the loss of SPINDOC results in reduced PARylation by PARP1 and the altered expression of genes known to be regulated by PARP1.

**SPINDOC facilitates PARP1-mediated PARylation in mice.** All the studies described above involve the in vitro manipulation of tissue-cultured cell lines. This begs the question: can SPINDOC also regulate PARP1 activity in vivo? Thus, to further investigate SPINDOC function, we engineered SPINDOC-null mice using a CRISPR-mediated knockout approach. Using the approach outlined in Fig. 7a, we generated three independent frame-shift-deletion alleles of SPINDOC (Fig. 7b). To mitigate issues that might arise from the off-target effects of CRISPR-mediated genomic manipulation, we selected two different lines (#18 and

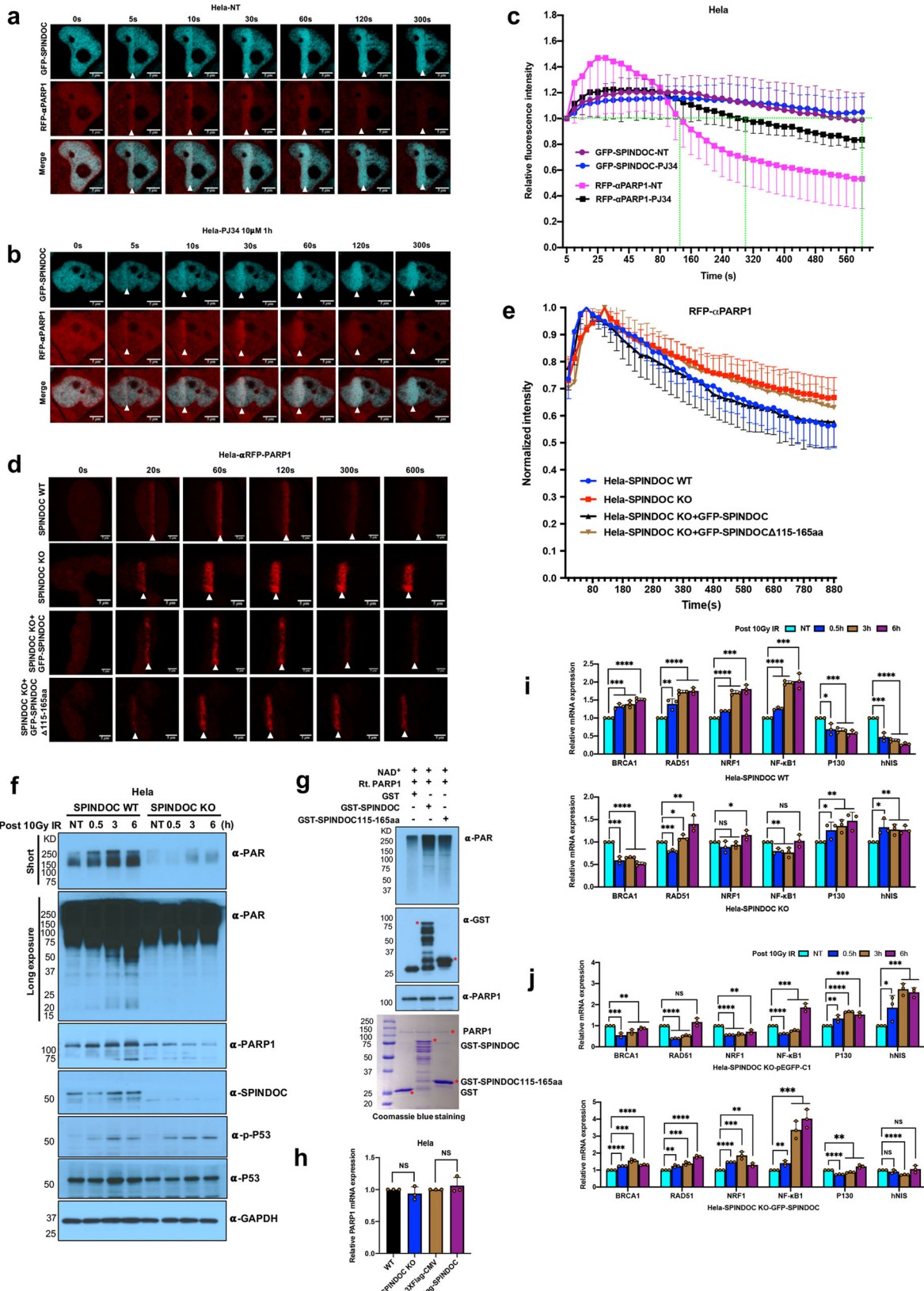

#19) and backcrossed them to the F3 generation before initiating our mouse studies (Fig. 7a). Importantly, SPINDOC KO mice are viable, 10–20% lighter than their WT littermates (Supplementary Fig. 7a), and the knockout mice display mendelian segregation (Supplementary Fig. 7b). Because PARP1-null mice are sensitive to IR treatment[12,13], we chose to subject a cohort of SPINDOC

KO and WT sibling control mice to irradiation with 6 Gy of X-ray, and then maintained these treated mice on antibiotics water and under sterile conditions. At three hours post IR, a cohort was sacrificed and thymi were taken for protein and mRNA extraction, the same numbers of nontreated KO and WT mice were sacrificed as controls for comparison. The thymus and intestine

**Fig. 6 SPINDOC facilitates PARP-mediated PARylation upon DSBs. a, b** Hela cells were cotransfected with GFP-SPINDOC and RFP-PARP1 nanobody plasmids, 24 h later, cells were nontreated (**a**), or treated with PJ34 10 μM an hour (**b**), before being subjected to 355 nm UVA laser microirradiation-coupled live-cell imaging. Cells were treated with 200 ng/ml Hoechst 30 mins for presensitization prior to laser damage and imaging. Confocal images (40X) were taken every 5 s for 10 mins and representative images of the irradiation-path signals are shown. **c** Quantification of the corecruitment of GFP-SPINDOC and RFP-αPARP1 in **a** and **b**. Graphs represent mean ± SD, $n = 10$ biologically independent cells. **d** Hela-WT cells, Hela-SPINDOC KO cells, and the same cells rescued with GFP-SPINDOC and GFP-SPINDOCΔ115–165aa mutant were transfected with RFP-αPARP1 nanobody plasmid and incubated for 24 h, before presensitizing with Hoechst (200 ng/ml), 355-nm UVA laser-induced damage and confocal imaging (40X). Images were taken every 20 s for 15 mins. **e** Quantification of RFP- αPARP1 recruitment in **d**. All timepoints are normalized to the peak of maximal recruitment of RFP-PARP1 to a value of 1.0 by each group. Graphs represent mean ± SD, $n = 10$ biologically independent cells. One-tailed Student $t$-tests were performed at 400 s, comparing Hela-WT with KO, $P = 0.0118$; Hela-KO vs KO-SPINDOC rescue, $P = 0.0480$; and Hela-KO $vs$ KO-SPINDOC mutant, $P = 0.0838$. **f** Hela-SPINDOC WT and KO cell lines were treated with 10 Gy IR, allowed to recover for 0.5, 3, and 6 h, and subjected to Western blot analysis using the indicated antibodies. The data are representative of three independent experiments. **g** An in vitro PARylation assay was performed in the presence of purified GST, GST-SPINDOC or GST-SPINDOC115–165 proteins, NAD$^+$, and recombinant PARP1, at room temperature for 30 min. Auto-PARylation signal was detected with αPAR antibody. The data are representative of three independent experiments. **h** PARP1 transcription level was evaluated by RT-qPCR in SPINDOC KO and overexpressed cells. One-tailed Student $t$-tests were performed, SPINDOC KO vs WT, $P = 0.1822$; Flag-SPINDOC vs vector, $P = 0.2270$. **i** Hela-SPINDOC WT and KO cell lines were treated with 10-Gy IR, allowed to recover for 0.5, 3, and 6 h, and subjected to RT-qPCR assay for the six indicated targets. Graphs represent mean ± SD, $n = 3$ biologically independent samples. One-tailed Student $t$-tests were performed, SPINDOC WT, **$P = 0.0044$, *$P = 0.0121$, ***$P < 0.001$, ****$P < 0.0001$. SPINDOC KO, RAD51: *$P = 0.0382$, **$P = 0.0091$. NRF1: *$P = 0.1027$, 0.0871, 0.0302. NF-κB1: $P = 0.0018$, 0.0093, 0.3797; P130: $P = 0.0304$, 0.0054, 0.0066; hNIS: $P = 0.0154$, 0.0068, 0.0030. **j** Hela-SPINDOC KO cells rescued with GFP-SPINDOC, pEGFP-C1 as a control, were treated with 10-Gy IR, allowed to recover for 0.5, 3, and 6 h, and subjected to RT-qPCR assay. Graphs represent mean ± SD, $n = 3$ biologically independent samples. One-tailed Student $t$-tests were performed, SPINDOC KO vector, BRCA1, $P = 0.0.0046$, 0.0043; NRF1: **$P = 0.0025$; P130: **$P = 0.0095$; hNIS: *$P = 0.0321$. SPINDOC KO-GFP-SPINDOC, RAD51: **$P = 0.0018$; NRF1: **$P = 0.0038$; NF-κB1: **$P = 0.0042$; P130: **$P = 0.0013$, 0.0080; ***$P < 0.001$, ****$P < 0.0001$, NS: nonsignificant.

are particularly sensitive to IR treatment, due to the high proliferation rates of cells within these tissues[48–50]. We chose to first investigate the impact of SPINDOC loss on the response to IR treatment in the thymus. Thymus extracts from nontreated SPINDOC KO mice, displayed a dramatic decrease in PARylation when compared with WT littermates, in both lines #18 and #19 (Fig. 7c, d). Upon IR treatment, PARylation within the thymic tissue of SPINDOC KO mice was also markedly reduced compared with WT, and PARP1 has undergone complete cleavage in line #18, and almost complete cleavage in line #19, which is indicative of IR-induced apoptosis within this tissue (Fig. 7c, d).

Consistent with in vitro cell experiments (Fig. 6i and Supplementary Fig. 6b), NF-κB mRNA expression in SPINDOC KO mice thymi was much reduced as compared with their WT littermates in both IR treatment and nontreatment samples, and this effect was seen in both KO lines. NF-κB mRNA levels in IR-treated WT mice were also clearly higher than nontreated WT mice, as expected for a PARP1-regulated target gene (Fig. 7e). On the contrary, the PARP1 negatively regulated gene P130 mRNA displayed the opposite effect (Fig. 7f). Moreover, when these mice were allowed to age, the survival curves of SPINDOC-deficient mice were significantly shorter than their WT siblings following IR exposure (Fig. 7g). Finally, histological analysis of the small intestine revealed increased cleaved Caspase-3 staining (a marker for apoptosis) in SPINDOC KO mice (Supplementary Fig. 7c, d), which likely explains the early lethality of these IR-treated KO mice. Thus, SPINDOC may play an important role in maintaining organismal homeostasis and survival in response to IR by promoting the PARP1-mediated PARylation of chromatin, and transcriptional regulators, and augmenting the positive and negative transcriptional-regulatory functions of PARP1.

## Discussion

Our initial intent was to expand our understanding of the SPIN1/SPINDOC protein complex, by identifying SPINDOC-interacting proteins. We hypothesized that SPINDOC is a scaffolding protein that recruits transcriptional coregulators to help SPIN1 control transcription. We did identify one such protein—PARP1, but surprisingly we found that SPINDOC interacted with PARP1 independent of SPIN1 (Fig. 1d, e). Interaction-mapping studies

show that there are distinct interaction domains in SPINDOC that mediate the binding with either PARP1 or SPIN1 (Fig. 2). We have proposed an interference model in which PARP1 binding to SPINDOC blocks the ability of SPIN1 to interact, and when a SPINDOC/SPIN1 complex is established, then PARP1 is prevented from associating with SPINDOC (Fig. 8). An analysis of the PhosphoSitePlus website reveals multiple phosphorylation sites with the two SPINDOC-interaction domains, which could regulate the selective interaction between PARP1 and SPIN1. We were not able to find evidence of trimeric SPINDOC/PARP1/SPIN1 complex by co-IP (Fig. 1d; PARP1 cannot co-IP SPIN1), and nor do all three proteins overlap in the isolated fractions from a glycerol gradient (Fig. 1e, except very weakly around fraction 15). However, it is possible that a very small amount of SPINDOC could be complexed with both PARP1 and SPIN1 (Fig. 8), which would prove to be a way for SPIN1 to recruit PAR activity to the promoter regions of the genes it regulates.

Different DNA lesions are recognized by the DNA-binding domain of PARP1 in similar conformations, helping to rationalize how this protein participates in multiple steps of DNA SSB and DSB repair. The DNA-binding domain of PARP1 interacts with DNA SSBs as a monomer through its second zinc finger (ZF)[51]. Upon DSB generation, PARP1 engages DNA as a monomer and adopts a compact conformation centered on the WGR domain. In this conformation, the WGR domain makes interdomain contacts with ZF1 and ZF3 and contributes to the formation of the DNA-binding interface[52]. We deleted each ZF found within the PARP1 DNA-binding domain and found that SPINDOC binding to PARP1 requires the fully intact DNA-binding domain, as even deletion of both ZF1and ZF3 does not block the SPINDOC interaction with PARP1 (Fig. 2a, b).

PARP1 enzymatic activity is regulated by DNA binding through its N-terminal ZF domains[14], and we propose that by interacting with the DNA-binding domain, SPINDOC may mimic the structural changes induced by DNA binding to promote PARP1's PARylated activity, as SPINDOC directly stimulates PARylation in vitro (Fig. 6g) and the loss of SPINDOC clearly reduces this activity in both cultured cells and in vivo (Fig. 6f, Supplementary Fig. 6a, Fig. 7c, 7d). This is not a novel concept. Indeed, a large number of PARP1-interacting proteins

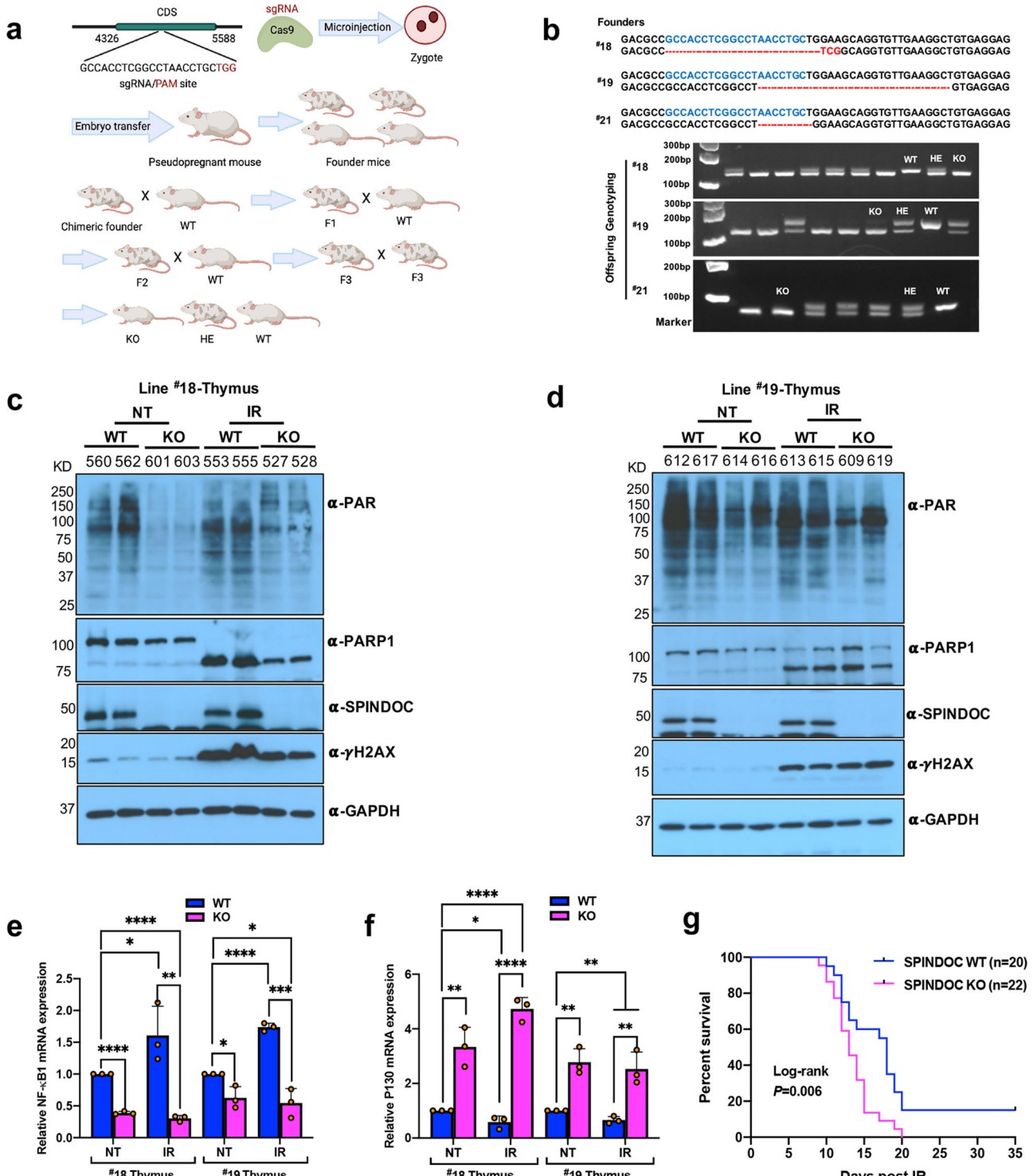

have been identified over the years[47,53]. A subset of these inter-acting proteins also binds the N terminus of PARP1 and pro-motes its activity, like SPINDOC does, and they include the RNA-binding proteins Sam68[54] and YB-1[55]. It should be noted that SPINDOC may also regulate the stability of PARP1, and not just its activity. In both the knockout HeLa (Fig. 6f) and 293T (Supplementary Fig. 6a) cell lines, we see this trend. Also, the lower level of PARP1 in SPINDOC KO mice (Fig. 7c, d), suggests that this regulation also occurs in vivo.

SPINDOC's interaction with PARP1 is associated with DDR, as first indicated from the analysis of RNA-seq data (Fig. 3a–c), and

confirmed by RT-qPCR and Western blot (Figs. 3d, S3c). In addition, the role of SPINDOC in DDR was further validated by its recruitment to laser-induced DNA-damaged stripes (Fig. 6a–c) and its requirement for the timely clearance of PARP1 from these stripes (Fig. 6d, e). Importantly, SPINDOC expression is also induced by DNA damage (both IR- and etoposide-induced) (Fig. 4), which has not been reported before. Furthermore, we show that this induction is mediated by KLF4 (Fig. 5). Interest-ingly, KLF4 itself is tightly regulated by PARylation[56]. Thus, the KLF4–SPINDOC–PARP1 axis may serve as a small-amplification loop that is engaged during DDR.

**Fig. 7 SPINDOC facilitates PARP1-mediated PARylation and targets transcription in vivo. a** Schematic of the establishment of SPINDOC KO mice. *Mus musculus* SPINDOC sgRNA is shown and Pam site is highlighted in red. Founders were backcrossed to the F3 generation before being intercrossed to generate KO mice. The model was created with BioRender.com. **b** Three independent founders were identified, harboring different out-of-frame indels (red) around the sgRNA sequence (blue). These founders were termed #18, #19, and #21. A PCR genotyping, using primers that straddled the edited regions, was developed. An example of the genotyping of a full litter from each of the three independent lines is shown. **c, d** SPINDOC KO from line #18 (**c**) and line #19 (**d**), and their corresponding WT littermates, were treated with 6-Gy IR and then allowed to recover for 3 h before their thymi were isolated. The thymi were lysed and subjected to Western blot analysis, using the indicated antibodies. The three-digit numbers above each lane refer to the identity number of the mouse from which the thymus was isolated. **e, f** Mice were treated as in **c, d** and their thymic tissue was sonicated before total RNA extraction and qRT-PCR analysis. Tissues were taken from sacrificed lines #18 and #19. Graphs represent mean ± SD, $n = 3$ biologically independent mice. One-tailed Student *t*-tests were performed. **e** Line #18, *$P = 0.0422$, **$P = 0.0041$; line #19, KO-NT vs WT-NT, $P = 0.0109$, KO-IR vs WT-NT, $P = 0.0127$. **f** Line #18, KO-NT vs WT-NT, $P = 0.0024$, WT-IR vs WT-NT, $P = 0.0161$; line #19, KO-NT vs WT-NT, $P = 0.0017$, KO-IR vs WT-IR, $P = 0.0035$, WT-IR vs WT-NT, $P = 0.0054$, KO-IR vs WT-NT, $P = 0.0066$. ***$P < 0.001$, ****$P < 0.0001$. **g** SPINDOC WT ($n = 20$) and KO ($n = 22$) mice were exposed to 6 Gy of IR and maintained for up to 35 days. Statistical significance of survival rates between genotypes was determined using the Kaplan–Meier method and *P* value was calculated by a log-rank test.

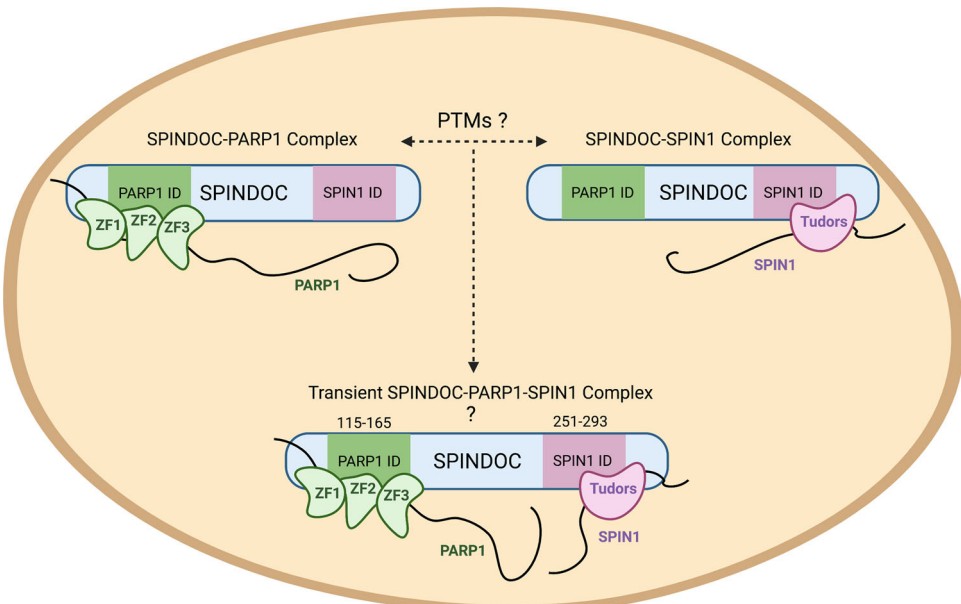

**Fig. 8 SPINDOC forms two stable protein complexes.** SPINDOC harbors a PARP1-interacting domain (ID) and a SPIN1 ID. Even though the two interacting domains are separated by ~100 amino acids, a stable complex of all three proteins is not observed, suggesting that the binding of PARP1 to SPINDOC sterically interferes with the SPIN1 interaction, and vice versa. It is unclear what regulates the inclusion of SPINDOC into the two different complexes, but the distribution is likely regulated by post-translational modifications (PTMs). Although we see no evidence of SPINDOC interacting with both its binding partners at the same time, it might be possible that a transient trimeric complex can form on rare occasions, which could possibly facilitate the recruitment of PARP1 to SPIN1 transcriptional targets. The model was created with BioRender.com.

SPINDOC is an understudied protein that was only recently functionally characterized as a SPIN1-interacting protein[6]. Here, we add to its biological functions, and show that it is a key regulator of PARP1 activity. SPINDOC has no enzymatic activity of its own, but it promotes PARylation by PARP1. Structural studies will be very valuable in elucidating these distinct properties of SPINDOC.

## Methods

**Cell culture**. HEK293T and Hela cell lines were purchased from ATCC. Cells were cultured in DMEM supplemented with 10% FBS, 1% penicillin/streptomycin, nonessential amino acids, and L-glutamine, and maintained in a humidified 37 °C incubator with 5% $CO_2$. All cell lines used in this study were routinely tested for mycoplasma by using MycoAlert™ (Lonza).

**Recombinant plasmid, siRNA synthesis, and transfection**. SPINDOC cDNA was inserted into pEGFP-C1 vectors to generate constructs expressing GFP-tagged SPINDOC in mammalian cells. pCMV-PARP1-3xFlag-WT was a gift from Thomas Muir (Addgene plasmid #111575)[57]. pcDNA3.1-HA-KLF4 FL was a gift from Michael Ruppert (Addgene plasmid # 34593)[58]. PARP1 Chromobody®-TagRFP plasmid (RFP-αPARP1 nanobody) was purchased from Chromotek (Cat# XCR).

Truncated or mutated GFP-SPINDOC and Flag-PARP1 constructs were generated by a Q5® Site-Directed Mutagenesis Kit (New England BioLabs, E0554) and sequenced by the MDACC Science Park Molecular Biology Core. Recombinant plasmids were transfected into HEK293T or Hela cell lines using polyethylenimine (PEI) according to the instructions of the manufacturer for 24~48 h. siRNAs of KLF4, SP1, and STAT3 were synthesized by Integrated DNA Technologies (IDT) and transfected into Hela cells with Lipofectamine™ 2000 Transfection Reagent (Invitrogen, Cat# 11668019). siRNA sequences are shown in Supplementary Table 1.

**Immunoprecipitation (IP) and Western blot analysis**. Cells were lysed with a mild lysis buffer (20 mM Tris-HCL (pH8.0),150 mM NaCl, 1 mM EDTA, and 0.5% Nonidet P-40) containing protease-inhibitor cocktail (Roche, Cat# 04693124001) on ice for 30 mins, sonicated with 1 s on/2 s off for 10 s by 30% amplitude of Sonic dismembrator Model 50 (Fisher Scientific), and centrifuged with 12000 g for 10 mins. The supernatant was incubated with 2μg of primary antibody overnight rotation at 4 °C, and then added 25μl protein A/G Ultralink® Resin (Thermo Scientific, Cat# 53132) for an hour rotation at 4 °C. Beads were washed and boiled in 1xSDS buffer. For regular samples, cells or tissues were lysed with RIPA lysis buffer with protease-inhibitor cocktail, then sonicated, and centrifuged using the same procedure as for IP samples. Proteins were separated by SDS-PAGE, transferred onto a PVDF membrane, and then blocked in 5% nonfat milk in PBST (0.05% Tween 20 with 1xPBS) buffer. Subsequently, the PVDF membranes were

probed with the indicated primary and secondary antibodies, and the signal was detected using ECL™ detection reagents (GE Healthcare, RPN2209). All antibodies are listed in Supplementary Table 2.

**Mass spectrometry**. HEK293T cells were transfected with pEGFPC1, GFP-SPINDOC, and GFP-SPINDOC del251–293aa, 48 h later, cells were harvested and separated into cytoplasmic and nuclear fractions. Each fraction was immunoprecipitated by anti-GFP and subjected to MS analysis. Anti-SPIN1 IP was performed by cytoplasmic and nuclear fractions from SPINDOC KO and the corresponding WT cell lines. Products of anti-GFP IP and anti-SPIN1 IP were separated by SDS-PAGE, and the corresponding gels were cut and sent to the Proteomics Facility at UT Austin. Proteins were reduced with DTT and alkylated with iodoacetamide, then digested in-gel with trypsin and desalted with Millipore 0.6 μl C18 ZipTip pipette tips, and then run by LC–MS/MS on a Thermo Ultimate 3000 RSLCnano UPLC in-line with an Orbitrap Fusion Tribrid mass spectrometer. The data were collected over a 3s cycle time with FT MS and ion-trap MS/MS. Raw data were searched using Proteome Discoverer 2.2 via Sequest HT search engine using 10-ppm mass tolerance for the MS from the FT detector and 0.6 Da for MS/MS from the ion trap detector with fixed modification of carbamidomethylation of cysteine, and variable modifications of methionine oxidation, protein N-terminal acetylation, and protein N-terminal acetylation with Met loss. Validation with Proteome Software Scaffold 4.1 used a protein threshold of 99% confidence for 2 peptides at peptide threshold of 1% FDR. Normalized spectral counts were used to compare protein levels across samples.

**Glycerol-gradient sedimentation**. Cell lysates were from HEK293T cells and HEK293T cells transfected with GFP-SPINDOC. Nine different glycerol solutions from 15 to 35% were made using 100% glycerol and layered slowly and evenly, from high to low, by depositing 500 μl of each concentration into a 5-mL ultra-centrifuge tube; 500 μl of cell lysates were then added gently on top of this gradient. WX Ultra 80 (Thermo Scientific) was used to centrifuge gradients at 4 °C, at 20,0000 g for 18 h using a rotor AH-650. After centrifugation, 26 fractions were slightly separated by pipettor with 200 μl each and boiled in 5xSDS buffer before Western blot analysis.

**Generation of SPINDOC CRISPR/Cas9-knockout (KO) cell lines**. To select the candidate sgRNAs for SPINDOC (C11orf84) genome editing, we used an online CRISPR tool GuidePro v2.1.0 generated by Dr. Xu's Lab at MD Anderson Cancer Center (https://bioinformatics.mdanderson.org/apps/GuidePro). SPINDOC sgRNA sequences are shown in Supplementary Fig. 1a. pLentiCRISPRv.2 plasmid was developed by Feng Zhang (Addgene Cat# 52961)[59] and was digested with BsmBI. The three sgRNA oligonucleotides were cloned into the plasmid according to a previously described protocol[60]. Details are described in the Supplementary methods.

**RT-qPCR**. Total RNA was extracted from cells using the RNeasy® Mini Kit (Qiagen, Cat# 74106) and reverse-transcribed using a Superscript III First Strand Synthesis Kit (Invitrogen, Cat# 18080-051). qPCR was performed with the Applied Biosystems 7900HT RT-PCR instrument using iTaq Universal SYBR® Green Supermix (Bio-Rad, Cat# 1725121) with indicated primers. All primers used in this study were synthesized by IDT and gene expression was calculated following normalization to GAPDH or β-actin levels using the comparative cycle-threshold method and was shown as folds, relative to the expression of each gene in the control cells. All primer sequences are shown in Supplementary Table 1.

**Dual-luciferase assay**. The KLF4-binding motif in the promoter region of SPINDOC was identified by JASPAR (http://jaspar.genereg.net/). pcDNA3.1-HA-KLF4 was a gift from Michael Ruppert (Addgene Cat# 34593)[58]. SPINDOC promoter sequence was identified using the genome browser Ensembl (http://www.ensembl.org/index.html) and aligned with the UCSC genome browser. SPINDOC promoter sequence was amplified by genomic DNA PCR, and the product was cloned into the pGL3-basic vector (OMEGA Engineering Inc.). The pGL3-basic-SPINDOC promoter mutant plasmid was subcloned from pGL3-basic-SPINDOC promoter plasmid with a mutant primer of KLF-binding motif. pRL-CMV was a Renilla luciferase control reporter vector. Different combinations of these plasmids were transfected into HEK293T cells for 48 h and treated with or without 5-Gy IR. Cells were harvested and we performed dual luciferase assays using a Luminometer (Promega) with two injection formats according to the Dual-Luciferase® Reporter (DLR™) Assay System (Promega, E1910).

**Knockout mice and genotyping**. To conduct CRISPR-mediated genome editing in a mouse, the complex of Cas9 DNA endonuclease and an accompanying Mus musculus SPINDOC sgRNA (GCCACCTCGGCCTAACCTGC*TGG*) was generated by Horizon Inspired Cell Solutions. The Transgenic Animal Facility at Science Park generated founder knockout mice (FVB strain) by mouse zygote pronuclear injection (Fig. 7a). Genomic DNA from the resultant 22 pups was isolated and used for PCR genotyping (forward primer: GCTGGTTGGCTAGGATCTGA, reverse primer: TCAGCTTGGATGTCTTGCAG). PCR products were purified and

sequenced to identify heterozygous (HE) mice (#18, #19, and #21) as founders, and the edited indel regions are shown in Fig. 7b. Founder mice were crossed to wild-type (WT) FVB mice, F1 heterozygous mice were backcrossed to wild-type mice to generate F2 heterozygous mice, and F2 further backcrossed to wild-type mice to generate F3 heterozygous mice. F3 heterozygous mice were intercrossed to generate homozygous SPINDOC KO and WT mice (Fig. 7a) and the representative genotyping results are shown in Fig. 7b. Genomic DNA was isolated from mouse tails and analyzed by PCR for genotyping using the corresponding primers; sequences were shown in Supplementary Table 1. All mouse experiments were reviewed and approved by the Institutional Animal Care and Use Committee at MD Anderson Cancer Center (ACUF# 00001090-RN02).

**Mouse maintenance and IR treatment**. SPINDOC KO, HE, and WT mice of both sexes from each strain were maintained in an AAALAC-accredited facility in individually ventilated cages, on aspen chip bedding. Purina Irradiated Breeder Diet (Lab Diet Cat# 5058) was provided ad libitum, and acidified reverse-osmosis water was provided by an automated system. Cages were maintained in a room where temperature and humidity were 20–22 °C and 55%, respectively, with a 14-h light and 10-h dark cycle and a minimum of 12 air changes per hour.

SPINDOC HE mice were intercrossed and the resulting pups were weaned at 21 days, and kept at 5 mice per cage, according to sex. SPINDOC KO and WT mice were aged to 4–5 weeks prior to IR treatment. Mice were moved to the IR-treatment room and transferred into sterile treatment cages in the biosafety cabinet. After 6.0 Gy of irradiation by RS-2000 biological irradiator (Rad Source), they were returned to a biosafety hood and transferred into sterile cages with sterile feed, bedding, and water containing 50 mg/ml of Clavamox. Antibiotic water bottles were changed out every five days and cages were changed out every seven days. Mice were monitored daily for symptoms of sickness and were euthanized if they were moribund as per federal and institutional guidelines. All the mouse experiments performed in this study were approved by the IACUC at MDACC.

**Chromatin immunoprecipitation (ChIP)-qPCR**. For the ChIP-qPCR experiments, a confluent 10-cm-dish of HEK293T cells was either treated with IR 10 Gy or not (the control), and then cross-linked using 1% formaldehyde (final concentration) followed by quenching with 125 mM glycine (Sigma). The cells were then harvested after a cold PBS wash. Cell pellets were resuspended in SDS lysis buffer (1% SDS, 10 mM EDTA, and 50 mM Tris at pH 8.1, protease-inhibitor cocktail) and 1-ml lysates in a 15-ml tube were sonicated by Bioruptor Plus (Diagenode) at 45 cycles of 30 s on/30 s off. After sonication, the sample was diluted 1:1 with ChIP dilution buffer (0.1% SDS, 2 mM EDTA, 20 mM Tris at pH 8.1, 1% Triton X-100, 500 mM NaCl, and protease-inhibitor cocktail), and 1% of the sample was retained as the input control. Lysates were incubated with 2 μg of anti-KLF4 and 2 μg of normal rabbit anti-IgG, respectively, at 4 °C rotation overnight. The next day, DNA-bound immunocomplexes were pulled down using 30μl protein A/G Ultralink® Resin at 4 °C for 1 h and subjected to serial washes with buffers of low salt, high salt, LiCl, and TE. The cross-linked DNA was then eluted twice with 100 μl of elution buffer (1% SDS and 0.1 M NaHCO₃) for 15 mins at room temperature. Subsequently, the samples, including input, were incubated with 5 M NaCl at 65 °C overnight to reverse the cross-linking, and then treated with RNase A (Sigma) at 37 °C for 30 mins and Proteinase K (Sigma) at 45 °C for 2 h. The DNA was then purified using a QIAquick PCR Purification Kit (Qiagen) and amplified by qPCR (primers are listed in Supplementary Table 1). The percent input was divided by signals obtained from input samples, which represented the amount of chromatin used in the ChIP. % Input = 100*power (2, adjust Input-Ct IP), adjust input = Ct Input-6.644.

**RNA-seq and GSEA analysis**. Total RNA was isolated from confluent 10-cm dish of HEK293T WT and KO cell lines using a RNeasy® Mini Kit and submitted to the Science Park NGS Core of The University of Texas MD Anderson Cancer Center for library preparation and sequencing. DNase I-treated RNA samples were fragmented and tagged at paired ends for stranded library preparation using the TrueSeq Stranded mRNA Kit (Illumina). The mRNA-seq was performed on the HiSeq 3000 platform (Illumina) at 2 × 75-bp paired-end runs. The normalized read count was generated from built-in functions in DESeq2. The differential expression analysis was performed with a DESeq2 bioconductor R package using a cutoff of FDR q ≤ 0.05. Differentiated genes were analyzed by GSEA from Broad Institute (https://www.gsea-msigdb.org/gsea/index.jsp). For GO annotations, we utilized the biological process GO terms with MsigDB format (.gmt). Expression-dataset files (.txt) and phenotype-label files (.cls) were generated according to the file format described in the GSEA user guide and GSEA data-format guide.

**Live-cell microirradiation imaging**. Hela cells were plated in a 35-mm FluoroDish with a 0.17-mm coverslip bottom (World Precision Instruments, FD35–100) for 12–24 h and cotransfected using PEI with 0.5 μg of GFP-SPINDOC plasmid and 1 μg of RFP-αPARP1 nanobody. Media was changed 4~6 h after transfected cells and cells were cultured for another 24 h. Next, cells were treated with (or without) PJ34 (Selleckchem, Cat# S7300) 10μM an hour and then changed the media with 200 ng/mL Hoechst33342 (Thermo Scientific, Cat# 62249) for 30 mins prior to laser-stripe microirradiation and imaging. Presensitized cells were rinsed and the

media replaced with phenol-free FluoroBright DMEM. For WT, SPINDOC KO, and rescue experiment, Hela cells were transfected with 1 μg of RFP-αPARP1 nanobody or cotransfected with 1 μg of GFP-SPINDOC/GFP-SPINDOC Δ115–165aa for 24 h. Laser-scanning confocal microscopy was performed using a Zeiss LSM880 and 40X W (N.A. 1.2) C-apo objective. UVA laser microirradiation was performed via 355-nm UVA optically pumped semiconductor laser (Coherent, Genesis) and time-lapse images captured pre- and post irradiation as indicated. The MDACC Science Park Flow Cytometry and Cell Imaging Core assisted with the live-cell microirradiation and with the generation of the images. For the quantitative evaluation of the recruitment kinetics, fluorescence intensities of the irradiated region were collected over the time course by Fiji Image J software and normalized to the preirradiation value, collecting fluorescence intensities of blank region as background. Corrected total cell fluorescence (CTCF) = Integrated Density - (Area of selected region X Mean fluorescence of background readings) (https://theolb.readthedocs.io).

**In vitro PARylation assay**. GST-SPINDOC and recombinant PARP1 (Abcam, Cat# ab123834) was subjected to in vitro PARylation at room temperature for 30 min in a reaction buffer (50 mM Tris-HCl, pH 8, 25 mM MgCl$_2$, and 50 mM NaCl) supplemented with 250 μM NAD$^+$. The reaction was stopped by adding 5 × SDS/PAGE sample buffer and the samples were analyzed by Western blot.

**Statistical analysis**. All quantitative experiments were carried out in triplicate, and graphs represent average ± standard deviation (SD). Statistical analysis was performed using unpaired, one-tailed or two-tailed Student's $t$-test for all experiments. Significance was determined; $P < 0.05$ (*) is considered to be significant and $P < 0.01$ (**), $P < 0.001$ (***), and $P < 0.0001$ (****) are considered to be highly significant. For GSEA analysis of RNA-seq data, significantly enriched pathways between genotypes were determined with a cutoff FDR value of q < 0.25. For statistical analysis of the mice-survival curve, a Kaplan–Meier estimate was generated and analyzed for statistical significance using a log-rank test.

**Reporting summary**. Further information on research design is available in the Nature Research Reporting Summary linked to this article.

## Data availability

The RNA-seq data generated in this study have been deposited in the GEO database under accession code GSE167306. All data supporting the findings of this study are available within the article and its supplementary information files. Additional information and relevant data will be available from the corresponding author upon reasonable request. Source data are provided with this paper. The source data contain raw data of blots/gels and reported averages in graphs and charts, underlying main figures Fig. 1d, e; Fig. 2b, d; Fig. 3d; Fig. 4a–c; Fig. 5b–f; Fig. 6c, e, f–h; Fig. 7b–g and Supplementary Fig. 1b, c; Supplementary Fig. 2a–e; Supplementary Fig. 3a–c; Supplementary Fig. 4a–c; Supplementary Fig. 5a, b; Supplementary Fig. 6a–f; Supplementary Fig. 7a, c. Source data are provided with this paper.

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

## Acknowledgements

M.T.B. is supported by CPRIT (RP180804). J.J.S. is supported by CPRIT (RP170002). C.J. and the SP-FCCIC is supported by CPRIT (RP170628). We thank Rebecca Deen for editing this paper. We thank Wei He for sgRNA design help and Rongjie Fu for help generating the CRISPR-mediated KO cell lines. We thank Michelle V. Gadush for assistance with the mass spectrometry studies.

## Author contributions

F.Y. performed most experiments, analyzed data, and wrote the draft. J.C. generated HEK293T-SPINDOC KO cell lines. B.L. analyzed RNA-seq data and submitted to GEO database. G.G. designed SPINDOC-KO mice-generating procedure. M.S. analyzed IHC data. C.J. assisted confocal image acquisition. J.S. assisted with designing the RNA-seq experiments. M.D.P. analyzed MS data. M.T.B supervised the project and wrote the final paper.

## Competing interests

M.T.B. is a cofounder of EpiCypher. All other authors declare no competing interests.
