## [Peer Review File · Nature Communications]

SPINDOC binds PARP1 to facilitate PARylationREVIEWER COMMENTS

Reviewer #1 (Remarks to the Author):

The manuscript by Yang et al reports a new interaction between SPINDOC and PARP1, which is independent from its previously reported connection with SPIN1. The authors identified the regions within SPINDOC and PARP1, which mediate their interaction. They generated SPINDOC knock-out in HEK293T and Hela cells and performed RNA-seq analysis of HEK293T KO vs WT, which revealed deregulation of different genes, including those involved in DNA damage response. Furthermore, the authors found that SPINDOC is upregulated in response to DNA damage (ionizing radiation and etoposide) and that the transcription factor KLF4 contributes to this regulation. SPINDOC is recruited to DNA damage sites and promotes PAR synthesis in cells and in mice.

I have listed below all my comments as they appear in the text. I have several major issues with the data and conclusions presented in the current manuscript:

- 1) The RNA-seq experiment to compare WT and SPINDOC knock-outs was performed without spike-in normalization, without which it is impossible to deduce the degree of transcriptional changes.
- 2) The rescue experiments are completely missing – to claim that the loss of SPINDOC is directly responsible for the observed transcriptional changes, the authors should do a complementation experiment by re-expressing SPINDOC in the KO.
- 3) To differentiate between PARP1 and SPIN1 complex-induced transcriptional changes, the authors should perform the complementation experiment not just with WT SPINDOC but also SPINDOC mutants: delta251-293 (no interaction with SPIN1) and delta115-165 (no interaction with PARP1).
- 4) Given the induction of SPINDOC after DNA damage and its effects on PAR levels, RNA-seq after DNA damage would be very informative.
- 5) It remains unclear why PARP1 persists at DNA damage sites: is there more DNA damage in SPINDOC KO, is there a defect in DNA damage signalling and DSB repair, is SPINDOC required for PARP1 dissociation by binding to its DBD (and outcompeting DNA) and can SPINDOC delta115-165 rescue this or not? If not, this would argue for a direct effect of SPINDOC on PARP1, if yes, then PARP1 recruitment may be affected indirectly. Do SPINDOC KO cells show reduced survival after DNA damage?
- 6) It seems that SPINDOC regulates PARP1 expression levels, not just PARP activity (Figure 6c) – is this a transcriptional regulation? Does SPINDOC regulate PARP1 activity directly or indirectly? This can be addressed with in vitro experiments with purified proteins.
- 7) The observation that genes like BRCA1 and RAD51 are deregulated in SPINDOC KO cells does not indicate that this is due to loss of PARP1 activity, as these genes are regulated by many other factors – the effect seen on these genes cannot be explained as a direct effect on PARP1. Can PARP1 overexpression rescue these effects in SPINDOC KO?

Other comments and suggestions:

Introduction:

- 1) Second paragraph is better suited at the end of the introduction as it brings forth new findings from this study. The authors can use just one explanatory sentence to transition into the paragraph on PARylation and incorporate the rest into the last paragraph of the introduction.
- 2) Recent studies have revealed that serines, and not lysines, are the major substrates of PARylation on histones. Please look into and reference the following articles: PMID 28190768, 29480802
- 3) There are many more examples of transcriptional regulation imparted by PARP1, please check the following reviews on this subject and update this section accordingly: PMID 28202539, 32029452

Results:

- 4) Figure 1f: co-expression of FLAG-PARP1 with GFP-SPINDOC results in increased protein levels of PARP1 based on FLAG signal in input and IP. This result does not seem to be reproducible as in Figure 2b PARP1 levels don't change in the input when GFP-SPINDOC is co-transfected. Please clarify this and please show an IP+WB experiment without overexpressing PARP1.
- 5) Figure S3: do Hela SPINDOC KO cells also show a growth phenotype?
- 6) Figure 3: GSEA analysis should be represented in the standard form of a bar graph with different categories on the y-axis (e.g., G2M checkpoint, UV response...) and log10 FDR on the x-axis. All

categories that are enriched among upregulated or downregulated genes should be presented, not just a small selection as shown in Fig. 3c and Fig. S3.

7) Figure 3: It would be informative to compare transcriptional changes in HEK293 and HeLa SPINDOC knock-outs to see whether the same genes involved in DDR are up/downregulated in HeLa also. RT-qPCR analysis would suffice.

8) In many anti-SPINDOC blots there is an additional lower band. Please clarify if this is another isoform or a degradation product? This band is, however, missing in Figure 4a for the IR treatment blot.

9) Figure 6a,b: the recruitment of PARP1 and SPINDOC should be properly quantified on a larger number of cells and presented in a graph showing recruitment kinetics. This would allow the authors to determine whether PARP1 or SPINDOC shows faster recruitment and would enable better appreciation of PARP1 retention in SPINDOC KO.

Reviewer #2 (Remarks to the Author):

The manuscript from the lab of Mark Bedford addresses a novel and interesting biochemical complex between PARP1 and SPINDOC, a Tudor-domain containing protein normally involved in regulating gene activity through interaction with SPIN1. The data identify SPINDOC as a protein that impacts cellular PARylation levels and whose expression is regulated by DNA damage signals. The protein recruits to laser-induced DNA damage sites and may interact directly with the PARP1 protein.

The paper adds nicely to the timely question of what proteins impact PARP1 activity, extending on the recent interest in HPF1 and other PARP interactors/modulators. While this is overall an interesting manuscript, a few areas worthy of (additional) experimental attention would strengthen the manuscript further: (a) do SPINDOC and PARP1 interact directly with each other or is the interaction bridged by DNA/nucleosomes, (b) do SPINDOC and PARP1 already exist as a complex in the nucleoplasm or do they only become a complex at DNA damage sites / to what extent do they exactly recruit together and with the same fast kinetics as PARP1 is known for? and (c) does SPINDOC mediate a direct impact on the biochemical PARylation activity of PARP1, as the excellent *in vivo* data might indicate?

Suggested experimental revisions

Figure 1:

This referee found the panels 1a and 1d not strictly necessary, since they reflect a fairly standard and established biochemical enrichment procedure followed by MS analysis. What would be useful and could benefit from further explanation, is how the MS results were plotted in Figure 1b and 1c. A protein whose spectral counts are similar in both the GFP-SPINDOC IP and the pEGFPC1 control cell would be expected to run diagonally ("the 45-degree line", as referred to by the authors), while proteins that specifically are detected only in the experimental sample (GFP-SPINDOC), but not in the negative GFP-only control (pEGFPC1) lie on the vertical Y-axis, as they have a spectral count of 0 in the control sample. In this context, figure 1c makes sense. Both SPINDOC fragment and PARP1 were found in the SPINDOC-fragment IP, but not in the GFP-only control, thus the points are found on the Y-axis with a X-axis spectral count of 0. SPIN1, in contrast, is detected in both SPINDOC and control IPs. This could make sense. However, in figure 1b, the IP'ed protein SPINDOC and its interactor PARP1 are pretty close to the diagonal line. Doesn't this indicate that both SPINDOC and control IPs contained similar amounts of SPINDOC and PARP1 peptides in the respective IPs? Why are a significant amount of PARP1 and SPINDOC identified by MS in the GFP-only control? This ought to be at least commented on and the figure legends should provide a bit more context. The western blot, in contrast, makes much more immediate sense to this referee (Figure 1f).

Further, could be authors comment on the other DNA-damage relevant PARP protein PARP2? This is not discussed. Since PARP2 has a very distinct DNA-binding domain from PARP1, it would be both useful and interesting to address whether and how SPINDOC interaction may be indeed specific for PARP1 and not for PARP2.

Figures 2 and S2

The mapping experiments indicate robustly that the PARP1 DBD is necessary for SPINDOC interaction. On the SPINDOC protein, the region encompassing residues 115-165 are necessary and sufficient for the interaction by co-IP and western blot. Using DNase I assays, the authors observed a stronger western blot signal in the DNase I-treated sample and thus conclude that SPINDOC and DNA may compete for binding to PARP1's DBD module. This conclusion appears premature. PARP1's DBD can be readily expressed and purified in heterologous systems and a fragment of SPINDOC 115-165 can likely be readily obtained as a fusion protein, e.g. GFP-, GST- or MBP-SPINDOC (115-165). The authors should test for this interaction directly and compete with addition of DNA in trans or, at a minimum, also use DNA intercalators to test in their cellular IPs whether they observe a stronger interaction between PARP1 DBD and SPINDOC (115-165).

Figures 3: the transcriptome profiling looks fine. It would be best to validate the up- and down-regulation of a few DDR components by western blot, as changes in steady-state mRNA levels may not provide a complete gene expression picture on the impact of SPINDOC deletion on the DDR pathway components.

Figure 4: The changes in SPINDOC expression level are interesting. Could the authors conduct IPs of SPINDOC in etoposide-treated cells and check whether this leads to changes (increase/decrease) in PARP1 interaction? Alternatively, any other experiment that could link the changed expression levels of SPINDOC to an altered interaction with / activity of PARP1 would be interesting.

Figure 6: these experiments test the interesting possibility that SPINDOC recruits to laser-induced DNA damage sites together with PARP1 and checks on the impact of a SPINDOC-deletion on PARP1 recruitment. The images appear to show that PARP1 recruits transiently (as expected), while SPINDOC recruits more for the long-term. However, there are a few problems with the current data: (a) the recruitment should be quantified and plotted as recruitment curves over time (since individual nuclei may behave differently; thus, there needs to be a sufficiently rigorous quantitation over many n's). (b) The kinetics of SPINDOC recruitment are longer and possibly also slower at the beginning (though this is hard to tell with only the 30s timepoint being shown). For PARP1, 30s is pretty late already, so it would be useful to collect also several earlier timepoints in order to establish whether the SPINDOC recruitment is together with PARP1 (which the author's data may imply) or whether SPINDOC rather might only bind to the laser-induced DNA damage sites once PARP1 has recruited and remodelled the chromatin there (note: many factors recruit to laser-induced DNA damage sites, but often only on the timescale of a minute or two (as shown for SPINDOC here). In many cases, this reflects not recruitment together with PARP1, but rather is a secondary effect to PARP1/2-induced chromatin remodeling. (c) The authors should repeat this experiment in the presence of PARP and (separately) PARP inhibitors. (d) For the benefit of green/red color-blind readers, please kindly use different colors, as it may be impossible for some readers to assess the co-recruitment of SPINDOC and PARP1 based on the presented data. Thank you.

Figure 7: very interesting data indicating changes in PAR levels both in physiological conditions, as well as upon IR treatment. PARP1 levels appear to be generally a bit lower in the SPINDOC KO mice. Could the authors comment on this a bit more? Also, if SPINDOC is a positive regulator of PARylation and thus presumably of PARP1 activity, it would be experimentally very interesting, if the authors could test this directly in vitro.

Short manuscript comments:

- should "Western blot" be lowercase?

Reviewer #3 (Remarks to the Author):

In this manuscript, Yang et al conduct a very thorough and insightful study yielding insights into a biological function of SPINDOC. Here they provide strong evidence that SPINDOC associates with PARP1 and plays a role in PARP1 mediated PARylation. The work begins with a proteomics analysis identifying SPINDOC and PARP1 as interactors which is supported by gel filtration chromatography, for example. Next the authors map the sites of association between SPINDOC and PARP1 using deletion analysis. RNA Seq comparing wild type and knockout SPINDOC cells reveals changes in genes associated with the DNA damage response, and they demonstrate that SPINDOC expression is induced by double strand breaks. Next they demonstrate that SPINDOC facilitates PARylation by PARP in Hela cells and in a knockout mouse. Importantly, SPINDOC knockout mice are more susceptible to ionizing radiation than wild type mice providing clear evidence for the importance of SPINDOC in the DNA damage response. Although this manuscript does not quite get to the precise mechanism of action of SPINDOC in the DNA damage response, this manuscript clearly makes the case for the importance of SPINDOC and links it strongly to being involved in PARylation by PARP1.

After having read this manuscript several times and reviewing all the evidence presented this is one of the very rare times where I feel the manuscript is acceptable for publication in its current form, although there may be very minor corrections needed in the writing that I missed.

REVIEWER REBUTTAL

We would like to thank the reviewers for their constructive criticism. The first two reviewers had a number of specific suggestions for improvement, and the third reviewer felt that the manuscript was acceptable for publication in its original form. We have addressed all the issues raised by the firsts two reviewers, and we now hope that they find this study suitable for publication. All changes to the text are marked in “green”. At the suggestions of the reviewers, we have modified a number of Figures. Here is a brief summary of the figure changes that we have made. All these changes are also specifically addressed in the detailed rebuttal below.

Changes to Figures:

Fig.1 Deleted **a** and **d**, remaining 5 panels were re-numbered. **b** was repeated and replaced, **c** and **e** were replaced by original nuclear MS peptide counts, and not presented as Log2.

Fig.S1 no change.

Fig.2 **b** was repeated and replaced.

Fig.S2 added **b-e**.

Fig.3 added **d**.

Fig.S3 added **b** and **c**.

Fig.4 **a** SPINDOC left panel was replaced, **c** graphs were modified to include data points.

Fig.S4 added **b** and **c**.

Fig.5 **c**, **e**, **f** graphs were modified to include data points and colors altered for colorblind readers.

Fig.S5 **a**, **b** graphs were modified to include data points and colors altered for colorblind readers, **a** key was removed.

Fig.6 the original **a** is replaced by **a-c**, the original **b** is replaced by **d** and **e**, we added **g**, **h**, **j**. **i** was modified to include data points and colors altered for colorblind readers.

Fig.S6 added **c-f**. **b** was modified to include data points and colors altered for the colorblind.

Fig.7 **b**, **e-g** were modified to include data points and colors altered for colorblind readers.

FigS7 **c** was modified to include data points.

Fig.8 no change.

Reviewer #1 (Remarks to the Author):

1) The RNA-seq experiment to compare WT and SPINDOC knock-outs was performed without spike-in normalization, without which it is impossible to deduce the degree of transcriptional changes.

We agree that the spike-in in RNA-seq will help the normalization across samples, especially in the case that gene expression levels experience globally changes, including housekeeping genes. However, the validation of the normalization in DE analysis without biased differential changes from a large number of genes can be confirmed with a set of housekeeping genes as well (ref 1,2), in which case that the spike-in could potentially be optional. Also, according to the PubMed database, there are many reference publications that used RNA-seq technology without spike-in (ref 3-7). The RNA-seq datasets in those references have shown the differential expressed gene signatures relating to DNA damage/repair. Such data mining suggests it is practical to perform an RNA-seq without spike-in if no globally biased gene expression happens, and if detailed validation is performed.

Reference:

1. Brief Bioinform. 2018 Sep; 19(5): 776–792. Selecting between-sample RNA-Seq normalization methods from the perspective of their assumptions.
2. Brief Bioinform. 2013 Nov;14(6):671-83. A comprehensive evaluation of normalization methods for Illumina high-throughput RNA sequencing data analysis.
3. Nat Commun. 2019; 10: 3143. BRCA2 abrogation triggers innate immune responses potentiated by treatment with PARP inhibitors.
4. Nat Commun. 2019; 10: 2792. Common and distinct transcriptional signatures of mammalian embryonic lethality
5. Nat Commun. 2014 Dec 19; 5: 5812. Genome-wide analysis of the human p53 transcriptional network unveils a lncRNA tumour suppressor signature
6. Nat Commun. 2021; 12: 1826. Mutant ASXL1 induces age-related expansion of phenotypic hematopoietic stem cells through activation of Akt/mTOR pathway
7. Nat Commun. 2019; 10: 3694. Priming mobilization of hair follicle stem cells triggers permanent loss of regeneration after alkylating chemotherapy

Importantly, we have gone on to validate the all the targets we highlighted in **Fig. 3b**. First by qPCR, in new **Fig. 3d** (including 4 genes that are not altered, as controls), but also by Western blot analysis to show that the RNA levels are a good indicator of the changes in protein levels as well (new **Fig. S3c**). Also, the protein level changes were shown in in both HeLa and 293T (WT vs KO) cells (new **Fig. S3c**).

2) The rescue experiments are completely missing – to claim that the loss of SPINDOC is directly responsible for the observed transcriptional changes, the authors should do a complementation experiment by re-expressing SPINDOC in the KO.

We have now performed the suggested complementation experiment. We have expanded **Fig. 6j** to now include a second panel with the rescue of SPINDOC KO cells by GFP-SPINDOC, in HeLa cells. We also performed the rescue in HEK293T cells (see **Fig. S6c**).

3) To differentiate between PARP1 and SPIN1 complex-induced transcriptional changes, the authors should perform the complementation experiment not just with WT SPINDOC but also SPINDOC mutants: delta251-293 (no interaction with SPIN1) and delta115-165 (no interaction with PARP1).

We have now performed this suggested complementation experiment using WT, SPINDOC mutants, in SPINDOC KO HeLa cells. This new data is shown in **Figure S6d**.

4) Given the induction of SPINDOC after DNA damage and its effects on PAR levels, RNA-seq after DNA damage would be very informative.

We have tested transcriptional changes of a panel of 6 PARP1 targets and performed SPINDOC rescue experiments as suggested from question #2 to address transcriptional changes after DNA damage. We do agree with the reviewer that additional data would be obtained from another RNA-seq experiment, and we are planning to do these experiments in the future.

5) It remains unclear why PARP1 persists at DNA damage sites: is there more DNA damage in SPINDOC KO, is there a defect in DNA damage signaling and DSB repair, is SPINDOC required for PARP1 dissociation by binding to its DBD (and outcompeting DNA) and can SPINDOC delta115-165 rescue this or not? If not, this would argue for a direct effect of SPINDOC on PARP1, if yes, then PARP1 recruitment may be affected indirectly. Do SPINDOC KO cells show reduced survival after DNA damage?

We have now performed the suggested rescue experiment. SPINDOC KO cells were rescued with GFP-SPINDOC or with GFP-SPINDOCdelta115-165. The delayed dissociation of PARP1,

after DNA damage, observed in the SPINDOC KO cells, is indeed rescued by GFP-SPINDOC, but not by GFP-SPINDOCdelta115-165. We have added both the images for this experiment (new **Fig.6d**) and the quantification of PARP1 dissociation as a consequence of the rescue with these two different vectors (new **Fig. 6e**).

We do not think that there is more DNA damage in SPINDOC KO cells that have not be challenged with a DNA-damaging agent, because there is no activation of p53 (phosphorylation) in the absence of SPINDOC (**Fig. 6f & Fig. S6a**)

Regarding the second part of the question, SPINDOC KO cells do display a significant reduced in survival after DNA damage, when compared to WT cells (new **Fig. S6f**).

6) It seems that SPINDOC regulates PARP1 expression levels, not just PARP activity (Figure 6c) – is this a transcriptional regulation? Does SPINDOC regulate PARP1 activity directly or indirectly? This can be addressed with in vitro experiments with purified proteins.

At the suggestion of this reviewer, we have now further investigated how SPINDOC regulates PARP1 activity. We first tested the question of transcriptional regulation. We overexpressed SPINDOC in WT HeLa cells, and also tested SPINDOC KO cells, and then checked PARP1 RNA level changes by qPCR. No changes were observed, and we can conclude that SPINDOC does not regulated PARP1 at the transcriptional level. This data is now presented in new **Fig. 6h**.

We also tested the ability of SPINDOC to activate PARP1 activity *in vitro*, using recombinant proteins. Here, recombinant and active PARP1 was purchased and mixed with GST-SPINDOC, in the presence of NAD⁺. We observe that GST-SPINDOC but not GST, can promote auto-PARylation of PARP1 (new **Fig. 6g**). This indicates that PARP1 is activated though a direct interaction with SPINDOC.

7) The observation that genes like BRCA1 and RAD51 are deregulated in SPINDOC KO cells does not indicate that this is due to loss of PARP1 activity, as these genes are regulated by many other factors – the effect seen on these genes cannot be explained as a direct effect on PARP1. Can PARP1 overexpression rescue these effects in SPINDOC KO?

We performed the experiment suggested by the reviewer, and transfected SPINDOC KO cells with Flag-PARP1, and then evaluated the expression levels of the 6 reported PARP1 targets, including BRCA1 and RAD51 (new **Fig. S6e**). When comparing this data to the expression levels of the 6 genes depicted in **Fig. 6i & 6j**, we indeed see a partial rescue of the induction (and repression) of known PARP1 target genes. This indicates that the reduction of PARP1 activity (caused by SPINDOC loss) can be overcome, to some degree, by the overexpression of PARP1.

Other comments and suggestions:

Introduction:

1) Second paragraph is better suited at the end of the introduction as it brings forth new findings from this study. The authors can use just one explanatory sentence to transition into the paragraph on PARylation and incorporate the rest into the last paragraph of the introduction.

We would prefer to retain the introduction as it is. The second paragraph of the introduction does briefly introduce new findings, particularly the discovery of a SPINDOC/PARP1 interaction. This paragraph is needed to provide a reason for introducing PARP1 functions in the third paragraph.

2) Recent studies have revealed that serines, and not lysines, are the major substrates of PARylation on histones. Please look into and reference the following articles: PMID 28190768, 29480802

This is an important reference that we had missed. It has now been added to the introduction, along with an explanatory sentence:

“When PARP1 is complexed with histone PARylation factor 1 (HPF1), it is able to PARylate serine residues on histone²⁰.”

3) There are many more examples of transcriptional regulation imparted by PARP1, please check the following reviews on this subject and update this section accordingly: PMID 28202539, 32029452

Both these key references have now been added to the introduction section. The following sentence has also been added:

“PARP1 and its activity also regulate RNA processing, like alternative splicing, RNA modification, mRNA stability and mRNA translation^{28, 29}.”

Results:

4) Figure 1f: co-expression of FLAG-PARP1 with GFP-SPINDOC results in increased protein levels of PARP1 based on FLAG signal in input and IP. This result does not seem to be reproducible as in Figure 2b PARP1 levels don't change in the input when GFP-SPINDOC is co-transfected. Please clarify this and please show an IP+WB experiment without overexpressing PARP1.

We have addressed this issue in two ways. First, we repeated the experiment depicted in Figure 2b, and we do see the stabilization of Flag-PARP1 when it is co-expressed with GFP-SPINDOC. This new figure now replaces the original figure (new **Fig. 2b**). Second, we have also performed a focused experiment to directly look at the stabilization of PARP1, by co-expressing different amounts of Flag-SPINDOC. Endogenous PARP1 becomes more stable as Flag-SPINDOC levels increase. This data is presented in new **Fig. S2d**.

5) Figure S3: do HeLa SPINDOC KO cells also show a growth phenotype?

We investigated the growth of HeLa SPINDOC KO cells, and found that they do proliferate more slowly than their WT counterparts. This data is presented in new **Fig. S3b**.

6) Figure 3: GSEA analysis should be represented in the standard form of a bar graph with different categories on the y-axis (e.g., G2M checkpoint, UV response...) and log₁₀ FDR on the x-axis. All categories that are enriched among upregulated or downregulated genes should be presented, not just a small selection as shown in Fig. 3c and Fig. S3.

Gene Ontology (GO) analysis or pathway analysis can be presented by a bar graph with GO Terms or pathways as y-axis and -log₁₀(FDR) as x-axis. However, it is typical to use the running score diagrams for the Gene Set Enrichment Analysis (GSEA) (ref 1-5). Furthermore, **Table S4** presents all differentially expressed genes, and **Table S5** presents all enriched gene sets from the H collection (hallmarks gene sets) in MSigDB (ref 6-8).

References:

1. Hayman, T.J., Baro, M., MacNeil, T. *et al.* STING enhances cell death through regulation of reactive oxygen species and DNA damage. *Nat Commun* **12**, 2327 (2021).

2. Sanij, E., Hannan, K.M., Xuan, J. *et al.* CX-5461 activates the DNA damage response and demonstrates therapeutic efficacy in high-grade serous ovarian cancer. *Nat Commun* **11**, 2641 (2020)
3. Olah, M., Patrick, E., Villani, AC. *et al.* A transcriptomic atlas of aged human microglia. *Nat Commun* **9**, 539 (2018)
4. Mackenzie, K., Carroll, P., Martin, CA. *et al.* cGAS surveillance of micronuclei links genome instability to innate immunity. *Nature* **548**, 461–465 (2017).
5. Santos, M., Faryabi, R., Ergen, A. *et al.* DNA-damage-induced differentiation of leukaemic cells as an anti-cancer barrier. *Nature* **514**, 107–111 (2014).
6. Subramanian, A., Tamayo, P., *et al.* Gene set enrichment analysis: A knowledge-based approach for interpreting genome-wide expression profiles. *PNAS* **102** (43) 15545-15550
7. Liberzon, A., Subramanian, A., *et al.* Molecular signatures database (MSigDB) 3.0, *Bioinformatics*, Volume 27, Issue 12, Pages 1739–1740
8. Liberzon, A., Chet Birger, *et al.* The Molecular Signatures Database (MSigDB) hallmark gene set collection. *Cell Syst.* 2015 Dec 23; 1(6): 417–425.

7) Figure 3: It would be informative to compare transcriptional changes in HEK293 and Hela SPINDOC knock-outs to see whether the same genes involved in DDR are up/downregulated in Hela also. RT-qPCR analysis would suffice.

We have performed the suggested RT-qPCR, for DDR genes, in both HEK293 and Hela SPINDOC KO cells. For all the tested genes, the transcriptional response is in the same direction, when SPINDOC is lost. This data is presented in new **Fig. 3d**.

8) In many anti-SPINDOC blots there is an additional lower band. Please clarify if this is another isoform or a degradation product? This band is, however, missing in Figure 4a for the IR treatment blot.

The lower band is actually present in **Fig. 4a**. When we made the original figure, we accidentally cut that lower band out. We have remade the figure to include a larger panel that encompasses more of the blot (new **Fig. 4a**). The lower band is much weaker in the IR-treated sample, when compared to the Etop-treated sample, and we are not sure of the reason for this. We also do not know if the lower band is a SPINDOC splice variant or a degradation product.

9) Figure 6a,b: the recruitment of PARP1 and SPINDOC should be properly quantified on a larger number of cells and presented in a graph showing recruitment kinetics. This would allow the authors to determine whether PARP1 or SPINDOC shows faster recruitment and would enable better appreciation of PARP1 retention in SPINDOC KO.

We have now quantified the recruitment of PARP1 and SPINDOC, as suggested by the reviewer. We also quantified the recruitment of PARP1 in SPINDOC KO cells, and rescued KO cells. This data is presented in new **Fig. 6a-e**.

Reviewer #2 (Remarks to the Author):

1) Figure 1:

a) This referee found the panels 1a and 1d not strictly necessary, since they reflect a fairly standard and established biochemical enrichment procedure followed by MS analysis. b) What would be useful and could benefit from further explanation, is how the MS results were plotted in Figure 1b and 1c. A protein whose spectral counts are similar in both the GFP-SPINDOC IP and the pEGFPC1 control cell would be expected to run diagonally (“the 45-degree line”, as referred to by the authors), while proteins that specifically are detected only in the experimental sample (GFP-SPINDOC), but not in the negative GFP-only control (pEGFPC1) lie on the vertical Y-axis, as they have a spectral count of 0 in the control sample. In this context, figure 1c makes sense. Both SPINDOC fragment and PARP1 were found in the SPINDOC-fragment IP, but not in the GFP-only control, thus the points are found on the Y-axis with a X-axis spectral count of 0. SPIN1, in contrast, is detected in both SPINDOC and control IPs. This could make sense. However, in figure 1b, the IP’ed protein SPINDOC and its interactor PARP1 are pretty close to the diagonal line. Doesn’t this indicate that both SPINDOC and control IPs contained similar amounts of SPINDOC and PARP1 peptides in the respective IPs? Why are a significant amounts of PARP1 and SPINDOC identified by MS in the GFP-only control? This ought to be at least commented on and the figure legends should provide a bit more context. The western blot, in contrast, makes much more immediate sense to this referee (Figure 1f).

1a) We agree that the original panels 1a and 1d were not necessary. They have been removed.

1b) We also agree that the way we presented the mass spec data is not very clear. The main concern is with the data presented in original **Fig. 1b**. We thus repeated the mass spec experiment, and new data is presented in new **Fig 1a**. We also changed the way we presented the mass spec data from Log2 (in the original figure) to total spectral counts (new figure), which helps separate the points better on the plot.

Here is a summary of the “original” vs “new” data for the GFP and GFP-SPINDOC mass spec:
We focused on the counts from the nuclear fraction.

“Original”	– SPIN1	– 0 for GFP and 61 for GFP-SPINDOC
	– PARP1	– 9 for GFP and 67 for GFP-SPINDOC
	– SPINDOC	– 241 for GFP and 627 for GFP-SPINDOC

“New”

– SPIN1	– 0 for GFP and 83 for GFP-SPINDOC
– PARP1	– 20 for GFP and 574 for GFP-SPINDOC
– SPINDOC	– 316 for GFP and 908 for GFP-SPINDOC

The ratio of peptides identified in the two separate experiments is fairly similar. There is some PARP1 “stickiness” for GFP, but there is a 28-fold enrichment of PARP1 for GFP-SPINDOC.

However, what is concerning (and hard to explain) is that a lot of SPINDOC peptides are coming down with GFP. We do expect GFP-SPINDOC to pull-down some endogenous SPINDOC, because we have unpublished data showing that SPINDOC can homodimerize, but it is really not clear why GFP on its own, is binding so much SPINDOC.

In the text, we have now added the following sentence to highlight this issue:

“We note that GFP-SPINDOC can enrich for SPINDOC itself, possible due to homodimerization, but there is also some binding of the GFP control to endogenous SPINDOC.”

2) Further, could be authors comment on the other DNA-damage relevant PARP protein PARP2? This is not discussed. Since PARP2 has a very distinct DNA-binding domain from PARP1, it would be both useful and interesting to address whether and how SPINDOC interaction may be indeed specific for PARP1 and not for PARP2.

This is an important question, and we performed a co-IP experiment to investigate if PARP2 can interact with SPINDOC. It does not. We have added this data in new **Fig. S2e**.

3) Figures 2 and S2

The mapping experiments indicate robustly that the PARP1 DBD is necessary for SPINDOC interaction. On the SPINDOC protein, the region encompassing residues 115-165 are necessary and sufficient for the interaction by co-IP and western blot. Using DNase I assays, the authors observed a stronger western blot signal in the DNase I-treated sample and thus conclude that SPINDOC and DNA may compete for binding to PARP1’s DBD module. This conclusion appears premature. PARP1’s DBD can be readily expressed and purified in heterologous systems and a fragment of SPINDOC 115-165 can likely be readily obtained as a fusion protein, e.g. GFP-, GST- or MBP-SPINDOC (115-165). The authors should test for this interaction directly and compete with addition of DNA in trans or, at a minimum, also use DNA

intercalators to test in their cellular IPs whether they observe a stronger interaction between PARP1 DBD and SPINDOC (115-165).

We did indeed only perform one experiment to support the idea that DNA can block the PARP1/SPINDOC interaction, and that was using DNase I. At the suggestion of the reviewer, we have tested this hypothesis in two additional ways. First, we used EtBr as a DNA intercalator in a co-IP experiment, and we observe a stronger interaction under these conditions (new **Fig. S2b**). Second, we observe a direct interaction between recombinant PARP1 (purchased) and GST-SPINDOC. Furthermore, when DNA is added to this *in vitro* pull-down reaction, we see a strong inhibition of the interaction (new **Fig. S2c**). Thus, we now present three independent experiments to suggest that DNA can modulate the PARP1/SPINDOC interaction.

4) Figures 3: the transcriptome profiling looks fine. It would be best to validate the up- and down-regulation of a few DDR components by western blot, as changes in steady-state mRNA levels may not provide a complete gene expression picture on the impact of SPINDOC deletion on the DDR pathway components.

Fig. 3b listed 11 DDR genes that are differentially regulated between SPINDOC WT and KO cells. We were asked by Reviewer #1 to validate these finding by qPCR, which we did and is presented in new **Fig. 3d**. At the request of this reviewer, we also investigated the expression changes at the protein level. We did this in both HeLa and 293T WT and KO cells. We were able to find good antibodies to 10 of the 11 targets highlighted in **Fig 3b**, and validated in new **Fig 3d**. In general, the Western blot analysis correlated very well with the RNA-seq and qPCR data (see new **Fig.S3c**)

5) Figure 4: The changes in SPINDOC expression level are interesting. Could the authors conduct IPs of SPINDOC in etoposide-treated cells and check whether this leads to changes (increase/decrease) in PARP1 interaction? Alternatively, any other experiment that could link the changed expression levels of SPINDOC to an altered interaction with / activity of PARP1 would be interesting.

We performed IPs of SPINDOC and PARP1 in etoposide-treated cells. First, we looked at the endogenous interaction and show that after etoposide-treatment there is a stronger interaction between SPINDOC and PARP1 (new **Fig. S4b**). This is even more obvious when ectopically

GFP-SPINDOC is IPed, before and after etoposide-treatment. We observe a very strong interaction of GFP-SPINDOC with endogenous PARP1, after DNA damage (new **Fig. S4c**).

6) Figure 6: these experiments test the interesting possibility that SPINDOC recruits to laser-induced DNA damage sites together with PARP1 and checks on the impact of a SPINDOC-deletion on PARP1 recruitment. The images appear to show that PARP1 recruits transiently (as expected), while SPINDOC recruits more for the long-term. However, there are a few problems with the current data: (a) the recruitment should be quantified and plotted as recruitment curves over time (since individual nuclei may behave differently; thus, there needs to be a sufficiently rigorous quantitation over many n's). (b) The kinetics of SPINDOC recruitment are longer and possibly also slower at the beginning (though this is hard to tell with only the 30s timepoint being shown). For PARP1, 30s is pretty late already, so it would be useful to collect also several earlier timepoints in order to establish whether the SPINDOC recruitment is together with PARP1 (which the author's data may imply) or whether SPINDOC rather might only bind to the laser-induced DNA damage sites once PARP1 has recruited and remodelled the chromatin there (note: many factors recruit to laser-induced DNA damage sites, but often only on the timescale of a minute or two (as shown for SPINDOC here). In many cases, this reflects not recruitment together with PARP1, but rather is a secondary effect to PARP1/2-induced chromatin remodeling. (c) The authors should repeat this experiment in the presence of PARP and (separately) PARG inhibitors. (d) For the benefit of green/red color-blind readers, please kindly use different colors, as it may be impossible for some readers to assess the co-recruitment of SPINDOC and PARP1 based on the presented data. Thank you.

6a) At the suggestion of both Reviewers 1 & 2, we have now quantified the recruitment of PARP1 and SPINDOC, and plotted this data as curves over time. Please see new **Fig. 6c** and **Fig. 6e**.

6b) We have collected this recruitment data at earlier time points, as suggested by the reviewer. We now include 5s and 10s time points. Please see new **Fig. 6a** and **Fig. 6b**.

6c) We have repeated this experiment in the presence of a PARP1 inhibitor (PJ34). And we see an effect on PARP1 recruitment, but not much effect on SPINDOC recruitment dynamics. Please see new **Fig. 6b** and **Fig. 6c**.

6d) We have taken to heart this reviewers' concerns about green/red color-blind readers, and their ability to interpret the data we are presenting. Our IF experiments are now presented in red and cyan. In addition, we have altered the graph in every figure to colors that can be read by green/red color-blind readers.

7) Figure 7: a) very interesting data indicating changes in PAR levels both in physiological conditions, as well as upon IR treatment. PARP1 levels appear to be generally a bit lower in the SPINDOC KO mice. Could the authors comment on this a bit more? b) Also, if SPINDOC is a positive regulator of PARylation and thus presumably of PARP1 activity, it would be experimentally very interesting, if the authors could test this directly *in vitro*.

7a) Yes, SPINDOC may also regulate the stability of PARP1, and not just its activity. In both the knockout HeLa (**Fig. S3c, Fig. 6f**) and 293T (**Fig. S3c, Fig. S6a**) cell lines we see this trend. As the reviewer notes, this lower level of PARP1 in SPINDOC KO mice (**Fig. 7c & 7d**), suggests this regulation also occurs *in vivo*. We now comment on this observation in the discussion.

7b) This is an important issue that was also raised by Reviewer #1. We also tested the ability of SPINDOC to activate PARP1 activity *in vitro*, using recombinant proteins. Here, recombinant and active PARP1 was purchased and mixed with GST-SPINDOC, in the presence of NAD⁺. We observe that GST-SPINDOC but not GST, can promote auto-PARylation of PARP1 (new **Fig. 6g**). This indicates that PARP1 is activated through a direct interaction with SPINDOC.

REVIEWER COMMENTS

Reviewer #1 (Remarks to the Author):

I would like to thank the authors for a thorough revision. I have a few remaining remarks.

1) Regarding spike-in normalization, I agree that RT-qPCR validation with control genes is sufficient. However, different graphs should be normalized in the same way (e.g., with 'WT no damage' sample) such that transcriptional changes caused by KO/rescue conditions and by DNA damage can be visualized and statistically compared. This applies to graphs in Fig. 6i, 6j and Fig. S6b, S6c.

2) Regarding the effect of SPINDOC overexpression on PARP1 levels, Fig. 2d shows no consistent effect of GFP-SPINDOC overexpression on FLAG-PARP1. It seems that only full-length GFP-SPINDOC shows somewhat higher FLAG-PARP1 levels. Furthermore, the new experiment in Figure S2d shows that GFP-SPINDOC overexpression has no effect on endogenous PARP1 levels – please have a look at input PARP1. In the MS text you conclude that there is a minor effect, in the Rebuttal you say that endogenous PARP1 becomes more stable as Flag-SPINDOC levels increase. Based on the blot from one experiment I would conclude that there is no effect. If the authors are convinced that there is an effect, then this experiment should be repeated three times and quantified. Otherwise the data suggest that stabilization of FLAG-PARP1 is an artefact of its overexpression.

Line 214: remove 'which is consistent with SPINDOC overexpression models (Fig.S2d)' unless you perform a quantification based on a triplicate experiment.

3) Laser experiments in Fig. 6b: for GFP-SPINDOC recruitment it is not clear where the laser line was applied, as at 5s a big part of the nucleus is bleached and the signal increases not only in the line on the right side but also in a dot on the left side. Please indicate on all laser recruitment images where exactly the nucleus was damaged using arrows.

4) Figure 6g: GST-SPIINDOC is very impure, suggesting that there are many degradation products and/or crossreactive co-purified proteins. To exclude non-specific PARP activation it is necessary to test whether delta115-165 has an effect on PARP activity. Please also include Coomassie gels for in vitro assays.

5) Line 200: plated instead of planted; line 214: except for instead of with the except for; line 257 KLF4 instead of KLK4; line 285 remove 'and'; line 334 when compared

Reviewer #2 (Remarks to the Author):

This referee had specific concerns and made several experimental suggestions, which the authors have addressed in their revised manuscript.

We also appreciate the color-blind-friendly use of colors in the revised manuscript. Thank you.

Specific comments on the experimental revision:

Figure 1 – MS data

The new data included in the new Fig. 1a are improved. The remaining issue is acknowledged by the authors and while the result remains puzzling, at least it is mentioned and discussed. I would thus judge my points to have been adequately addressed.

PARP2

The new data indicate that indeed the interaction is specific for the PARP1 member of the nuclear poly-ART family of enzymes. The addition of this evidence in the core text of the manuscript will be of interest to the PARP community.

Figure 2

We had asked whether the interaction between PARP1 and SPINDOC mediated by PARP1'2 DNA-

binding domain was direct and could be competed by DNA. The new data (provided in Fig. S2b,c) address this question satisfactorily.

Figure 3

We asked whether the observed, modest changes in gene expression levels for several transcripts correlated at the protein level. Using multiple western blots, the authors have now addressed this question. At least for ATF1, CREBBP and PPP2R2C, a slight/modest increase is observable, which supports the author's interpretation. The new Fig. S3c is thus fine.

Figure 4

The new data indicating how DNA damage mediated by etoposide increases SPINDOC and PARP1 interaction (in Fig. S4b,c) is interesting and adds potential impact to the manuscript.

Figure 6

Our most significant concern related to the quantitation of the live-cell imaging results, which was missing in the original version of this manuscript and the authors have now sought to address. While the new data provide a clearer and statistically more reliable summary of the recruitment kinetics, which is what this referee was seeking in order to see how SPINDOC and PARP1 behave, I now have several significant concerns about the results and the interpretation.

- The authors argue that SPINDOC deletion alters the association/retention of PARP1 at the laser-induced DNA damage sites (Fig. 6e; compare open blue circles with red filled squares). There does seem to be a bit of a delay to the release of RFP-PARP1 from the damage sites. However, when they re-introduce WT SPINDOC, the authors claim that PARP releases quickly again. I am not convinced about this, since RFP-PARP1 recruitment is already much lower at 80-300 seconds, so the fact that its enrichment is lower later, e.g. at times >600 seconds, could be the result of decreased recruitment (see the lower maximal peak of recruitment at about 70 seconds) relative to the other conditions. My personal take on this is that the rescue experiment do not show at all what the authors interpret. The presented data are premature in my opinion.
- The PARP inhibitor PJ34 appears to almost completely abolish RFP-PARP1 recruitment (Fig. 6c), which the authors interpret as reduced and delayed. I fail to see any convincing recruitment in Fig. 6a for RFP-PARP1 in the PJ34 condition. Certainly this is not visible in Fig. 6b, the representative images for the data quantitated over many nuclei in Fig. 6c. Why would PJ35 treatment almost completely (Fig. 6c) or completely (Fig. 6b) abolish PARP1 recruitment? This is not expected in the field.
- The authors claim that PJ34 treatment impacts PARP1 recruitment, but that "SPINDOC recruitment and resident time was not affected by deficient PARP1 activity (PJ34-treatment)." (line 288/289 of the revised manuscript). To me it looks a little decreased and certainly not visible at all in the representative images shown in Fig. 6b. The GFP-SPINDOC recruitment in Fig. 6a and 6b are wholly different, so I am a little befuddled of the authors' interpretation.
- The general problem with the assays shown in Fig. 6a-e is that the overall, maximal recruitment observed on their microscope appears to be rather low for PARP1 and thus the results "more noisy" than is typically seen in the field. In addition, there appears to be significant bleaching of fluorescence, as can be seen by the fact that the fluorescence drops below 1.0 (or, in fact, one cannot see any fluorescence for RFP-PARP1 after about 300 seconds in panel 6a and 6b. Why is this?
- Further, the recruitment kinetics of SPINDOC vs. PARP1 appear to differ. While PARP1 recruitment appears to peak after approximately 25 seconds (as seen by others in the field), SPINDOC recruitment appears to peak after about 60-120 seconds, a much lower timescale, more typical of proteins that respond to secondary events at the DNA damage sites, which FOLLOW PARP1 recruitment and the beginnings of its (fast) release kinetics. This does not speak for a role of SPINDOC as a direct, biochemical interactor of PARP1, but rather argues that its recruitment to DDR sites follow PARP1. This is important, in our opinion, as the authors summarize their findings in the abstract by stating that SPINDOC is "recruited to DNA lesions together with PARP1". Based on the data presented in Figure 6, I would say that this is not so clear. The only way to make that claim would probably be through two approaches: (i) test SPINDOC recruitment dynamics in a PARP1 KO cell and (ii) improve on the

imaging issues (small overall recruitment strength/robustness and bleaching of RFP fluorophore).
- Overall, I find the data presented in Fig. 6a-e premature and not publishable in their present form.

Figure 7

This last point is addressed adequately.

We would like to thank the reviewers for their constructive suggestion, and the manuscript is now much improved. To aid with tracking, we have marked all the changes in “green” in the body of the text.

Reviewer #1 (Remarks to the Author):

I would like to thank the authors for a thorough revision. I have a few remaining remarks.

1) Regarding spike-in normalization, I agree that RT-qPCR validation with control genes is sufficient. However, different graphs should be normalized in the same way (e.g., with ‘WT no damage’ sample) such that transcriptional changes caused by KO/rescue conditions and by DNA damage can be visualized and statistically compared. This applies to graphs in Fig. 6i, 6j and Fig. S6b, S6c.

In Fig. 6i, 6j and Fig. S6b, S6c, RT-qPCR were performed to test the transcriptional changes of PARP1 targets upon DNA damage in SPINDOC WT, KO and rescued cell lines. We feel that it would be better to comparing IR treatment with non-treat in the same cell line. From current data, we can clearly see these targets response in a SPINDOC-dependent manner. We think that this is the simplest way to present this data. Thus, we would like to retain Fig. 6i, 6j, S6b, S6c, as they are.

2) Regarding the effect of SPINDOC overexpression on PARP1 levels, Fig. 2d shows no consistent effect of GFP-SPINDOC overexpression on FLAG-PARP1. It seems that only full-length GFP-SPINDOC shows somewhat higher FLAG-PARP1 levels. Furthermore, the new experiment in Figure S2d shows that GFP-SPINDOC overexpression has no effect on endogenous PARP1 levels – please have a look at input PARP1. In the MS text you conclude that there is a minor effect, in the Rebuttal you say that endogenous PARP1 becomes more stable as Flag-SPINDOC levels increase. Based on the blot from one experiment I would conclude that there is no effect. If the authors are convinced that there is an effect, then this experiment should be repeated three times and quantified. Otherwise the data suggest that stabilization of FLAG-PARP1 is an artefact of its overexpression.

Line 214: remove 'which is consistent with SPINDOC overexpression models (Fig.S2d)' unless you perform a quantification based on a triplicate experiment.

At the request of this reviewer, we have now removed the statement "which is consistent with SPINDOC overexpression models (Fig.S2d)" from the results section.

3) Laser experiments in Fig. 6b: for GFP-SPINDOC recruitment it is not clear where the laser line was applied, as at 5s a big part of the nucleus is bleached and the signal increases not only in the line on the right side but also in a dot on the left side. Please indicate on all laser recruitment images where exactly the nucleus was damaged using arrows.

At the request of this reviewer, we have added triangles to indicate the position of all laser stripes on the images. This has been done for Fig. 6a, b & d.

4) Figure 6g: GST-SPIINDOC is very impure, suggesting that there are many degradation products and/or crossreactive co-purified proteins. To exclude non-specific PARP activation it is necessary to test whether delta115-165 has an effect on PARP activity. Please also include Coomassie gels for in vitro assays.

This is an important control, which we had not included in the original figure. We purified GST, GST-SPINDOC and GST-SPINDOC115-165aa to perform the *in vitro* PARylation assay again, and found GST-SPINDOC115-165aa could activate the PARP1 activity as well, shown in new **Fig.6g**, and also including the Coomassie blue stained gel. We would regard this as important additional data that strengthens our findings.

We have added the following sentence in the text to address this addition "Furthermore, a GST fusion harboring only the region that interacts with PARP1 (GST-SPINDOC115-165) is sufficient to induce this activation". The figure legend has also been updated.

5) Line 200: plated instead of planted; line 214: except for instead of with the except for; line 257 KLF4 instead of KLK4; line 285 remove 'and'; line 334 when compared.

We have modified the text to include the corrections of all five points made here.

Reviewer #2 (Remarks to the Author):

This referee had specific concerns and made several experimental suggestions, which the authors have addressed in their revised manuscript. We also appreciate the color-blind-friendly use of colors in the revised manuscript. Thank you.

This reviewer was happy with all our changes and additional data, except for some of the new data presented in Figure 6. We have addressed these issues below:

Figure 6

Our most significant concern related to the quantitation of the live-cell imaging results, which was missing in the original version of this manuscript and the authors have now sought to address. While the new data provide a clearer and statistically more reliable summary of the recruitment kinetics, which is what this referee was seeking in order to see how SPINDOC and PARP1 behave, I now have several significant concerns about the results and the interpretation.

- The authors argue that SPINDOC deletion alters the association/retention of PARP1 at the laser-induced DNA damage sites (Fig. 6e; compare open blue circles with red filled squares). There does seem to be a bit of a delay to the release of RFP-PARP1 from the damage sites. However, when they re-introduce WT SPINDOC, the authors claim that PARP releases quickly again. I am not convinced about this, since RFP-PARP1 recruitment is already much lower at 80-300 seconds, so the fact that its enrichment is lower later, e.g. at times >600 seconds, could be the result of decreased recruitment (see the lower maximal peak of recruitment at about 70 seconds) relative to the other conditions. My personal take on this is that the rescue experiment do not show at all what the authors interpret. The presented data are premature in my opinion.

We now detail the time of release of α RFP-PARP1 from the damage sites, using green line to indicate when PARP1 hit the baseline of 1.0. From Fig.6e, α RFP-PARP1 recruitment reached its peak at 60~80s upon different SPINDOC protein levels (this is quite different from HeLa cells co-transfected with GFP-SPINDOC and α RFP-PARP1, in which SPINDOC overexpression may affect PARP1 recruitment). However, α RFP-PARP1 release time that spans the initiation of laser damage to descend to the baseline 1.0 is significantly different. In HeLa-SPINDOC WT cell

line, the release time is 440s, but it is 620s in KO cell line, indicating a delay to the release of α RFP-PARP1 upon SPINDOC KO. When KO cells rescued with GFP-SPINDOC, it gets back to 360s, even the peak does not reach as high as WT, but PARP1 does release quickly again. While KO cells rescued with GFP-SPINDOC Δ 115-165aa, the time is 760s, still delay, indicating that SPINDOC mutant does not rescue SPINDOC's function in this context. However, please note that Cas9 expression is present in the Hela SPINDOC KO cell line as it was generated using CRISPR/Cas9 technology. This may result in some degree of DDR activity in these KO cells, which could confound the interpretation of PARP1 recruitment dynamics. We have now mentioned this caveat in the text with the following statement:

“We should note here that Cas9 expression is present in the Hela SPINDOC KO cell line, as the cell line was generated using CRISPR/Cas9 technology. This may result in some degree of DDR activity in these KO cells, which could confound the interpretation of PARP1 recruitment dynamics presented here.”

- The PARP inhibitor PJ34 appears to almost completely abolish RFP-PARP1 recruitment (Fig. 6c), which the authors interpret as reduced and delayed. I fail to see any convincing recruitment in Fig. 6a/b for RFP-PARP1 in the PJ34 condition. Certainly, this is not visible in Fig. 6b, the representative images for the data quantitated over many nuclei in Fig. 6c. Why would PJ34 treatment almost completely (Fig. 6c) or completely (Fig. 6b) abolish PARP1 recruitment? This is not expected in the field.

To address this issue, we have replaced the image of the representative cell in Fig 6b. Now you can clearly see that α RFP-PARP1 is recruited in the PJ34 condition. At the suggestion of the 1st reviewer, we have also added arrows to indicate the position of the laser stripe.

Our data suggest that PJ34 reduces the recruitment of PARP1 to sites of DNA damage, but does not abolish the recruitment. We were unable to find any publication that did a quantified time-course of PARP1 recruitment, in the presence of PJ34. So, it is unclear if this phenomenon has been observed before. There are papers that show that PARP1 does go to sites of DNA damage in the presence of PJ34, but these studies never did the quantification experiments.

- The authors claim that PJ34 treatment impacts PARP1 recruitment, but that “SPINDOC recruitment and resident time was not affected by deficient PARP1 activity (PJ34-treatment).”

(line 288/289 of the revised manuscript). To me it looks a little decreased and certainly not visible at all in the representative images shown in Fig. 6b. The GFP-SPINDOC recruitment in Fig. 6a and 6b are wholly different, so I am a little befuddled of the authors' interpretation.

We have replaced the image of the representative cell in Fig 6b. Now you can clearly see that GFP-SPINDOC is recruited in the PJ34 condition. We do not see a striking effect of PJ34 on SPINDOC recruitment, and we would like to retain the statement - "SPINDOC recruitment and resident time was not affected by deficient PARP1 activity (PJ34-treatment)." - in the text.

- The general problem with the assays shown in Fig. 6a-e is that the overall, maximal recruitment observed on their microscope appears to be rather low for PARP1 and thus the results "more noisy" than is typically seen in the field. In addition, there appears to be significant bleaching of fluorescence, as can be seen by the fact that the fluorescence drops below 1.0 (or, in fact, one cannot see any fluorescence for RFP-PARP1 after about 300 seconds in panel 6a and 6b. Why is this? Are there reports that PARP1 is gone from sites of damage at 300 sec – need a ref, and address in text.

PARP1 reporter α RFP-PARP1 what we used in this study is a novel PARP1-affinity reagent, which is based on a PARP1-specific single-domain antibody fragment (~ 15 kDa), termed nanobody, which recognizes the N-terminus of endogenous PARP1 with nanomolar affinity (Ref.1). The properties of this nanobody have been described by the Rothbauer group. The signal of the PARP1 chromobody (commercial name) is only transiently present at the DNA repair sites. In Fig. 8 of the Rothbauer paper, they show a nice IF time-course that tracks the RFP signal after damage. They use 10s, 26s, 200s and 420s timepoints to track dynamic changes in this signal. Unfortunately, they do not have a 300s timepoint. However, they see signal loss between 200s and 420s, which is in keeping with there being a loss of α RFP-PARP1 at 300s, in our experiments.

To address this issue, we have added the following sentence to the text, as well as the new key reference: "PARP1 is cleared within 300s (**Fig. 6a & b**), which is in keeping with reports characterizing the α RFP-PARP1 nanobody that was used in this study⁴⁵."

Key Reference:

1. Buchfellner A, Yurlova L, Nüske S, Scholz AM, Bogner J, Ruf B, Zolghadr K, Drexler SE, Drexler GA, Girst S, Greubel C, Reindl J, Siebenwirth C, Romer T, Friedl AA, Rothbauer U., A New Nanobody-Based Biosensor to Study Endogenous PARP1 In Vitro and in Live Human Cells. PLoS One. 2016 Mar 7;11(3):e0151041. doi: 10.1371/journal.pone.0151041. eCollection 2016.PMID: 26950694.

- Further, the recruitment kinetics of SPINDOC vs. PARP1 appear to differ. While PARP1 recruitment appears to peak after approximately 25 seconds (as seen by others in the field), SPINDOC recruitment appears to peak after about 60-120 seconds, a much lower timescale, more typical of proteins that respond to secondary events at the DNA damage sites, which FOLLOW PARP1 recruitment and the beginnings of its (fast) release kinetics. This does not speak for a role of SPINDOC as a direct, biochemical interactor of PARP1, but rather argues that its recruitment to DDR sites follow PARP1. This is important, in our opinion, as the authors summarize their findings in the abstract by stating that SPINDOC is “recruited to DNA lesions together with PARP1”. Based on the data presented in Figure 6, I would say that this is not so clear. The only way to make that claim would probably be through two approaches: (i) test SPINDOC recruitment dynamics in a PARP1 KO cell and (ii) improve on the imaging issues (small overall recruitment strength/robustness and bleaching of RFP fluorophore).

The reviewer is indeed correct, and the recruitment dynamics (after damage) of PARP1 and SPINDOC differ. We do know that PARP1 and SPINDOC do interact directly (all the data in Fig. 1 & 2). However, we do not know if this happens at sites of DNA damage. Indeed, we know that the same region of PARP1 interacts with both DNA and SPINDOC, and that excess DNA will inhibit the PARP1/SPINDOC interaction (Fig. S2a-c). So, it is possible that PARP1 is initially recruited to sites of DNA damage by a DNA interaction, and subsequently SPINDOC is somehow recruited by binding the same region of PARP1. How this potential hand-off may occur is unclear, and would be an important avenue of research in the future. In the abstract, we have now modified our original statement “recruited to DNA lesions together with PARP1”, as follows: “recruited to DNA lesions with dynamics that follows PARP1”.

We do agree with the reviewer that the IF experiments depicted in Fig. 6a-e do raise a number of additional questions, which we plan to address in future studies. Here, we have now replaced the representative images for Fig.6b (with a better image). We have also made a number of

additions/changes to the text to more accurately reflect the data. We hope that our study is now in a position to be accepted for publication.

REVIEWER COMMENTS

Reviewer #1 (Remarks to the Author):

Without using the same sample for normalization, it is not possible to compare WT and KO effects. Please include plots where all samples were normalized against 'WT no damage' in your Rebuttal, which will also be published and will allow readers to evaluate differences between WT and KO as well (and not just after DNA damage). Thank you.

Reviewer #2 (Remarks to the Author):

The authors have responded to my main concerns, as follows:

A) Figure 6e:

I was concerned in the initial and revised versions of the manuscript that the RFP-PARP1 recruitment was lower at 80-300 seconds in the context of the GFP-SPINDOC rescues in the SPINDOC KO cells and that this obfuscated interpretations of release times. The GFP-SPINDOC Delta 115-165aa mutant, however, appears 'delayed', yet again this may be complicated by the fact that the maximally observed recruitment of RFP-PARP1 at around 80-100 seconds is higher than for the wild-type GFP-SPINDOC.

The authors have now sought to improve the data analysis by detailing the time of release with a green line. This is helpful, but does not dispel my concern.

Do the authors still observe a full rescue with the GFP-SPINDOC and a delay with the Delta 115-165aa mutant if they normalize the peak of maximal recruitment of RFP-PARP1 to a value of 1.0 for all conditions tested. This is typically done in the field if direct side-by-side comparisons and the effect of an inhibitor or mutant is sought, as the authors have done in the present manuscript. See for example: Fig. 5d in Zandarashvili et al., 2020, *Science*, 10.1126/science.aax6367; or Fig. 4 in Juhasz et al., 2020, *Science Advances*, 10.1126/sciadv.abb8626.

I would love to see these normalized data to be convinced of the current interpretation, which I believe remains insufficiently supported.

B) Figure 6a,b,c. RFP-PARP1 recruitment in the PJ34 condition:

The newly provided images for RFP-PARP1 recruitment under these conditions is now more in-line with what is seen with other PARP inhibitors.

C) My befuddlement about GFP-SPINDOC in Figs. 6a and 6b.

This is now addressed by the new data.

D) Question related to the more "noisy" RFP-PARP1 recruitment

The provided new sentence and literature reference is very useful and addresses my concern.

E) Different recruitment kinetics between SPINDOC and PARP1

The authors have now acknowledged my concern by adapting the original statement to reflect the apparent delay of SPINDOC maximal enrichment relative to the earlier PARP1 recruitment. While this of course raises more questions that do somewhat impact one of the central findings of this study, I

agree that this can be left at the moment and can be the subject of further studies.

Reviewer #1 (Remarks to the Author):

Without using the same sample for normalization, it is not possible to compare WT and KO effects. Please include plots where all samples were normalized against 'WT no damage' in your Rebuttal, which will also be published and will allow readers to evaluate differences between WT and KO as well (and not just after DNA damage). Thank you.

We have done the normalization of HeLa and HEK-293T cell lines harboring SPINDOC WT and KO with or without IR treatment. Now readers will be easy to compare WT and KO effects upon DNA damage (see Fig.1, related to Fig.6i&j in the main manuscript). At the reviewers' request, the figure is included in the rebuttal, but not placed in the main manuscript.

Fig.1 Transcriptional changes of PARP1 targets normalized to SPINDOC “WT no DNA damage”.

HeLa (A) and HEK-293T (B) cell lines harboring SPINDOC WT, KO treated with 10 Gy IR and followed by 0.5h recovery. RT-qPCR was performed to evaluate transcriptional changes of PARP1 targets upon DNA damage, all groups are normalized to SPINDOC WT non-treated (NT) with DNA damage. * $P < 0.05$, ** $P < 0.01$, *** $P < 0.001$, **** $P < 0.0001$, NS, non-significant.

Reviewer #2 (Remarks to the Author):

The authors have responded to my main concerns, as follows:

A) Figure 6e:

I was concerned in the initial and revised versions of the manuscript that the RFP-PARP1 recruitment was lower at 80-300 seconds in the context of the GFP-SPINDOC rescues in the SPINDOC KO cells and that this obfuscated interpretations of release times. The GFP-SPINDOC Delta 115-165aa mutant, however, appears 'delayed', yet again this may be complicated by the fact that the maximally observed recruitment of RFP-PARP1 at around 80-100 seconds is higher than for the wild-type GFP-SPINDOC.

The authors have now sought to improve the data analysis by detailing the time of release with a green line. This is helpful, but does not dispel my concern.

Do the authors still observe a full rescue with the GFP-SPINDOC and a delay with the Delta 115-165aa mutant if they normalize the peak of maximal recruitment of RFP-PARP1 to a value of 1.0 for all conditions tested. This is typically done in the field if direct side-by-side comparisons and the effect of an inhibitor or mutant is sought, as the authors have done in the present manuscript. See for example: Fig. 5d in Zandarashvili et al., 2020, Science, 10.1126/science.aax6367; or Fig. 4 in Juhasz et al., 2020, Science Advances, 10.1126/sciadv.abb8626.

I would love to see these normalized data to be convinced of the current interpretation, which I believe remains insufficiently supported.

We have re-made the Fig.6e referring to the above two published papers as the reviewer suggested. Now we indeed observe a full rescue with the GFP-SPINDOC and a delay with the Delta 115-165aa mutant. For the rebuttal Fig.2 (main Fig.6e), time courses are normalized to its peak of maximal recruitment of RFP- α PARP1 to a value of 1.0. In SPINDOC WT cell line, RFP- α PARP1 reached a peak at 60s, and at 400s, the signal came back to the background level. However, in SPINDOC KO cell line, a peak is reached at 120s, and the clearance time is about 600s. When KO cells are rescued with GFP-SPINDOC, both the recruitment peak time (60s) and the clearance time (360s) are similar to the WT. However, when the KO cell lines were transfected with the SPINDOC mutant, both the recruitment peak time (120s) and the clearance time (660s) are similar to the KO. The data indicate that SPINDOC does rescue SPINDOC's function in this context, but SPINDOC mutant does not. We thank the reviewer for his/her patience with us regarding this figure, and all the very constructive suggestions for improved presentation.

Fig. 2 (new Fig. 6e in Manuscript) Quantification of RFP- α PARP1 recruitment.

All timepoints are normalized to the peak of maximal recruitment of RFP-PARP1 to a value of 1.0 by each group. Student t tests were performed at 400s, comparing HeLa-WT to the HeLa-KO, $P=0.012$; HeLa-KO vs KO-SPINDOC rescue, $P=0.048$; and HeLa-KO vs KO-SPINDOC mutant, $P=0.084$.

We are pleased that all the other concerns raised by this review, were satisfactorily addressed.